



# High-resolution inventory to capture glacier disintegration in the Austrian Silvretta

Andrea Fischer[1], Bernd Seiser[1], Kay Helfricht[1], Martin Stocker-Waldhuber[1]

[1] Institute for Interdisciplinary Mountain Research, Austrian Academy of Sciences, Innsbruck, Austria

*Correspondence to*: Andrea Fischer (andrea.fischer@oeaw.ac.at)

**Abstract.** Eastern Alpine glaciers have been receding since the LIA maximum, but the majority of glacier margins could be delineated unambiguously for the last Austrian glacier inventories. Even debris-covered termini, changes in slope, colour or the position of englacial streams enabled at least an in situ survey of glacier outlines. Today the outlines of totally debris-

covered glacier ice are fuzzy and raise the theoretical discussion if these glaciogenic features are still glaciers and should be part of the respective inventory – or part of an inventory of transient cryogenic landforms. A new high-resolution glacier inventory (area and surface elevation) was compiled for the years 2017 and 2018 to quantify glacier changes for the Austrian Silvretta region in full. Glacier outlines were mapped manually, based on orthophotos and elevation models and patterns of volume change of 1 to 0.5 m spatial resolution. The vertical accuracy of the DEMs generated from 6 to 8 LiDAR points per m² is in

the order of centimetres. calculated in relation to the previous inventories dating from 2004/2006 (LiDAR), 2002, 1969 (photogrammetry) and to the Little Ice Age maximum extent (moraines). Between 2004/06 and 2017/2018, the 46 glaciers of the Austrian Silvretta lost -29±4% of their area and now cover 13.1±0.4 km². This is only 32±2% of their LIA extent of 40.9±4.1 km². The area change rate increased from -0.6%/year (1969-2002) to -2.4%/year (2004/06-2017/18). The annual geodetic mass balance showed a loss increasing from -0.2±0.1 m w.e./year (1969-2002) to -0.8 m ±0.1 w.e./year (2004/06-

2017/18) with an interim peak in 2002-2004/06 at -1.5±0.7 m w.e./year. Identifying the glacier outlines offers a wide range of possible interpretations of former glaciers that have evolved into small and now totally debris-covered cryogenic geomorphological structures. Only the patterns and amounts of volume changes allow us to estimate the area of the buried glacier remnants. To keep track of the buried ice and its fate, and to distinguish increasing debris cover from ice loss, we recommend inventory repeat frequencies of three to five years and surface elevation data with a spatial resolution of one

metre.



# 1 Introduction

As a reaction to climate warming, mountain glaciers all over the world are changing at an increasing pace (IPCC, 2019) and
at unprecedented rates (Zemp et al., 2015), but with significant regional variability (IPCC, 2019). It is widely accepted that in
some mountain ranges a significant part of today's glaciers may disappear within this century (e.g. Zemp et al., 2019a, Huss
and Fischer, 2016a). A loss of almost all mountain glaciers by 2300 seems possible (Marzeion et al., 2012). For tackling
climate change using glaciers as essential climate variables (Bojinski et al., 2014), the precise and detailed monitoring of
glacier recession is essential (Zemp et al., 2019b). It can also reveal regionally different response times (Zekollari et al., 2020),
various uncertainties (Huss et al., 2014) and serve as a basis for specific scenarios (e.g. Zekollari et al., 2019).

Glacier inventories are amongst the most valuable tools for glacier monitoring on a regional scale (Haeberli et al., 2007, Gärtner
Roer et al., 2019), and for remote and large glaciers. After the pioneering World Glacier Inventory (WGI), compiled by WGMS
and NSIDC (1999), several initiatives, such as GLIMS (Kargel et al., 2014, Racoviteanu et al., 2009), have published global
glacier inventories like the Randolph glacier inventory (Pfeffer et al., 2014). International consortia like Globglacier work out
guidelines for mapping glaciers, e.g. Paul et al. (2009). At the same time, smaller regional studies work on new methods (e.g.
Paul et al, 2020) and have responded to regional phenomena and demands, for example, debris-covered glaciers (Nagai et al.,
2016) or the large and often cloudy and snow-covered Patagonian glaciers (Meier et al., 2018). Regional inventories at high
resolution can also serve as validation for large-scale semi-automatic remote sensing products.

Airborne LiDAR has been a valuable tool for glacier studies for nearly 20 years (e.g. Geist and Stötter, 2002, Pellikka and
Rees, 2009). Acquisition technologies, processing and analysis have been significantly enhanced since the early years to reach
a few cm nominal vertical accuracy in flat areas. High point densities allow processing gridded elevation data with a spatial
resolution of 0.5 m or even higher. Early scientific work included the investigation of the potential of (repeat) LiDAR data to
map geomorphological processes (Höfle and Rutzinger, 2011) and glacier/rock glacier extent (Abermann et al. 2010). In 2004,
federal authorities in Austria initiated the compilation of the first federal DEMs based on LiDAR which were used to update
the photogrammetric Austrian glacier inventories (Fischer et al., 2015). Now a repeat federal LiDAR DEM is available for
several regions in Austria.

In this article, we present a new LiDAR-based glacier inventory of the Austrian Silvretta (Figure 1), located in the federal
states of Tyrol and Vorarlberg. This inventory presents the glacier area and surface elevation for the years 2017 (Vorarlberg)
and 2018 (Tyrol). Here, for the first time, a regional glacier inventory was derived from two high-resolution LiDAR surveys
for a period of beginning glacier downwaste. Not only was it necessary to include the volume changes to delineate the debris-





covered glacier margins as proposed by Abermann et al. (2009), the volume change was used to estimate the geodetic mass balance of the 46 glaciers.

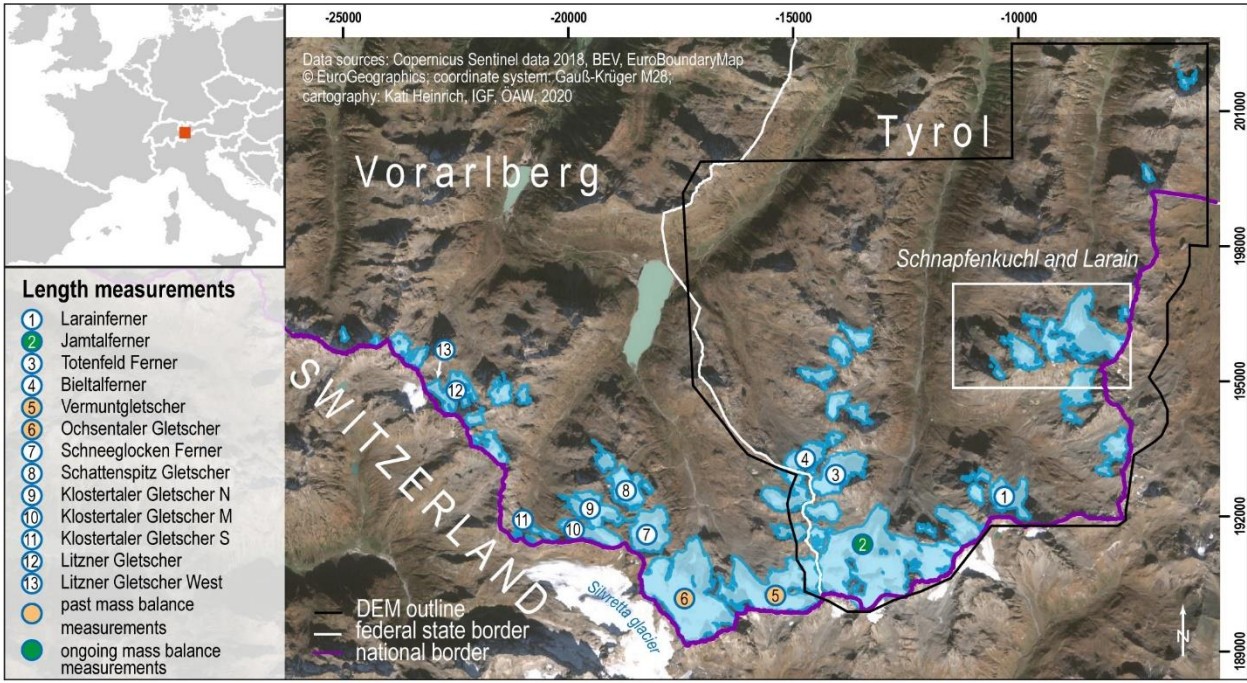

**Figure 1: Overview of the glaciers, ongoing and past mass balance and fluctuation time series in the Austrian part of the Silvretta range**

Heavy and continuous mass losses resulted in significant changes of glacier geomorphology and topology in the Austrian Silvretta, as illustrated in Figures 2 and 3 for the Schnapfenkuchl glaciers V, M and H and in Figure 4 for Litzner glacier. The direct mass balance of Jamtalferner has been strongly negative for the last two decades, with an ELA above summits for most years and several years with zero accumulation area. Beginning in the year 1892, annual length change measurements have been taken at the glaciers Jamtalferner, Southern Totenfeldferner, Bieltalferner, Vermunt Gletscher, Ochsentaler Gletscher, Schneeglocken Gletscher, and Klostertaler Gletscher M (Fischer et al., 2018, Fischer et al., 2016b). Some time series, however, have been abandoned: at Larainferner in 1993 (dead ice body at the undefined terminus), at Schattenspitz Gletscher (debris cover) and Klostertaler Gletscher S in 1995, at Klostertaler Gletscher N in 2003 (danger of rock fall) and at Litzner Gletscher in 2013 (debris cover all over the terminus).

The three Schnapfenkuchl glaciers as they present themselves today (Figure 2) cannot at first glance be identified even during a field survey, as bare ice is rarely visible (Figure 3), and the geomorphological structure of the surface is not dominated by ice dynamics as could be expected for lager debris-covered valley glaciers.



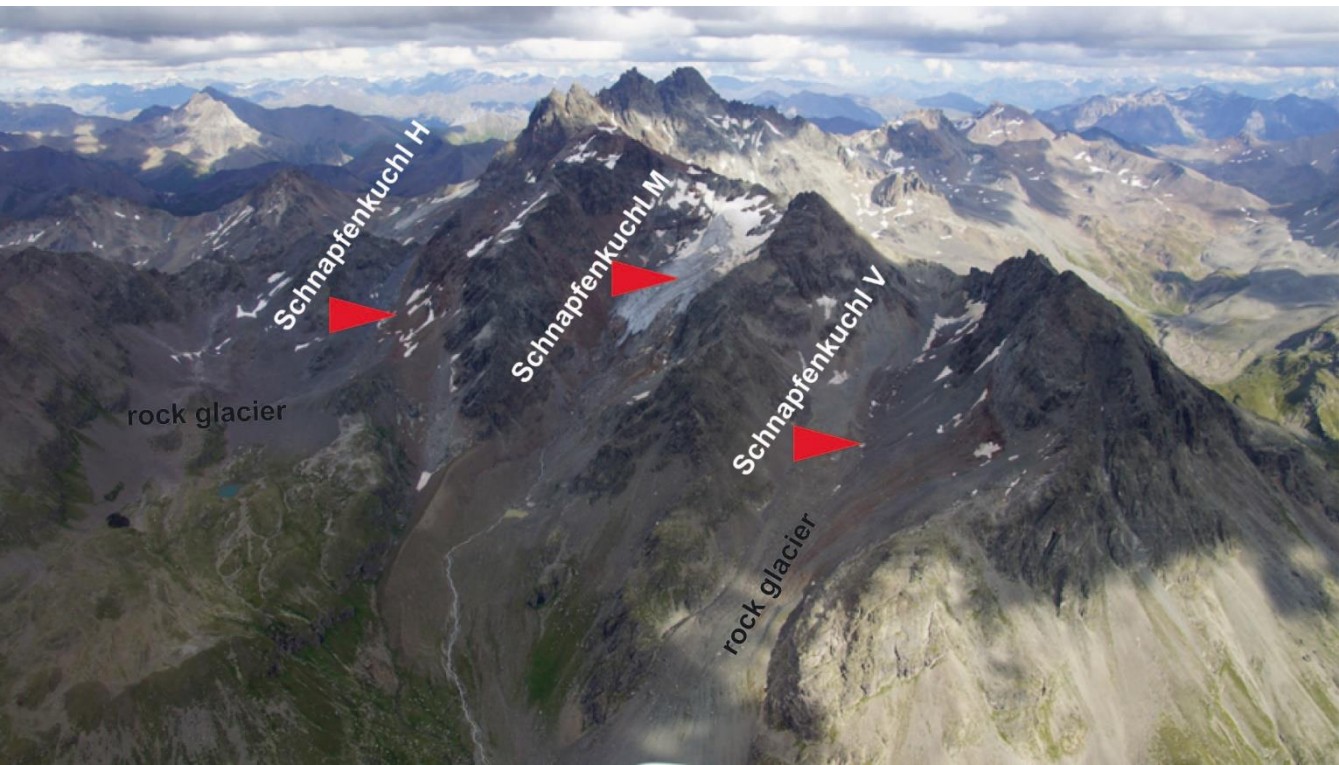

**Figure 2:** Overview of the Schnapfenkuchl glaciers V, M, and H in an aerial photograph of 21.08.2020 (photographer Andrea Fischer), which is typical for the current state for the glaciers of Austrian Silvretta studied in this paper: The glaciers are small, with a minimised accumulation area and increased debris cover, so that bare ice is rarely exposed. This raises the question if these transient cryogenic landforms are still glaciers and how we can monitor at which point the glaciers can actually be defined as 'gone'.





**Figure 3: Bare ice exposed at Schnapfenkuchl glacier V in the orthophoto of 2015 (red arrows), with stratigraphic layers (yellow arrows) indicating sedimentary ice or firn. Orthophotos: CC 4.0 https://www.data.gv.at/katalog/en/dataset/orthofoto**


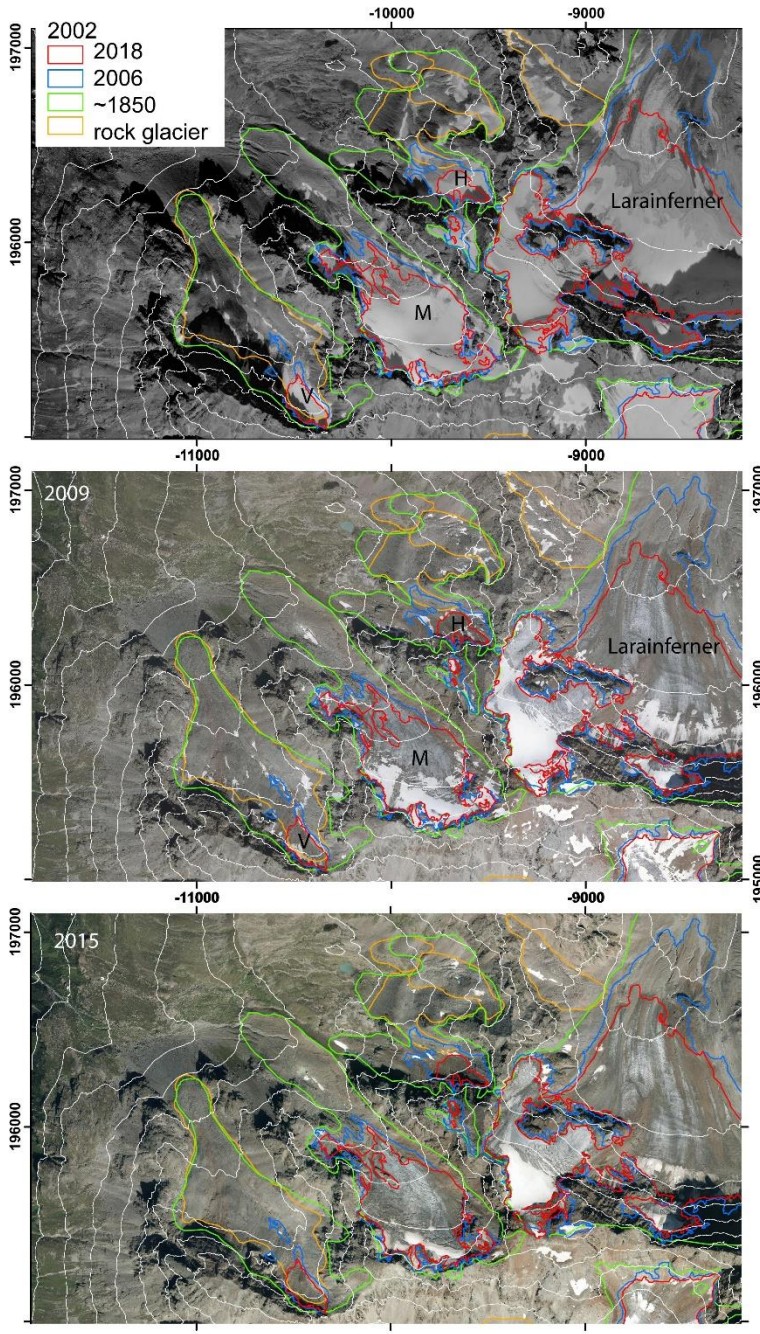

**Figure 4: The orthophotos of 2002, 2009 and 2015 of the Schnapfenkuchl glaciers/Larainferner show that the glacier changed quite rapidly within this time. The former accumulation area of Larainferner lost contact with the tongue between 2002 and 2009, last remnants of permanently visible glacier ice at eastern Schnapfenkuchl glacier became covered with debris between 2009 and 2019. Orthophotos: CC 4.0 https://www.data.gv.at/katalog/en/dataset/orthofoto. Rock glacier outlines: Krainer and Ribis (2012).**



We analyse Schnapfenkuchl V and H as if they were glaciers, because these structures were clearly identified as glaciers in past inventories: exposed bare ice, accumulation areas, an englacial drainage system and crevasses as indicators for ice dynamics were present in 2006 and before (Figure 4). In recent years these traditional and evident properties of a glacier became hidden by debris. In the orthophotos of 2002, 2009 and 2015 it is evident that Schnapfenkuchl glaciers could be clearly

identified as glaciers in 2002, while in 2009 and 2015 we would hardly map any glaciers at these locations.

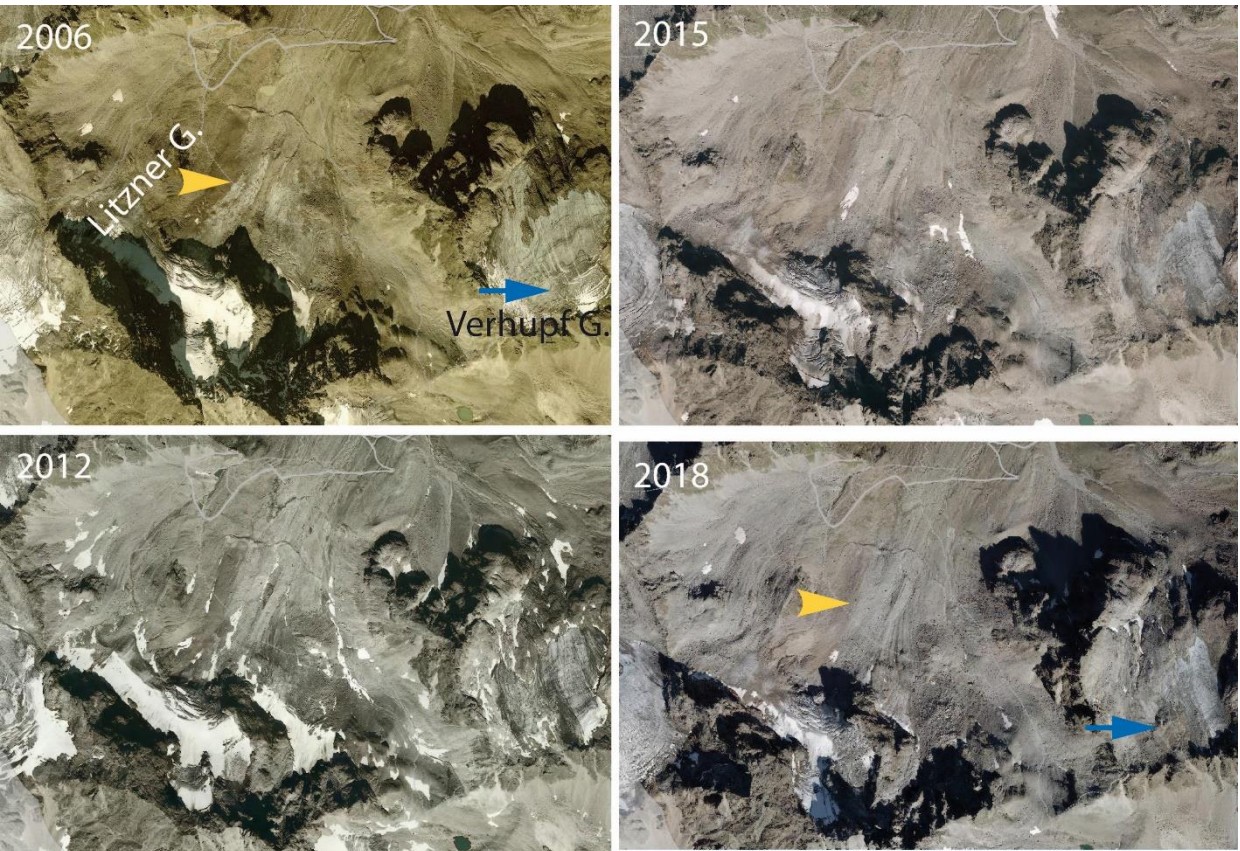

**Figure 5: Time series of orthophotos of Litzner glacier. Between 2006 and 2018, we lose track of glacier ice at the tongue (yellow arrow). The flat summit glaciers (blue arrow), intact in 2006, completely disintegrates as melt rates of 1.5 m close to the summits lead to a total loss of the shallow ice, leaving behind rock outcrops and debris-covered ice in small**

**bedrock troughs. Orthophotos: CC 4.0 https://www.data.gv.at/katalog/en/dataset/orthofoto**

The increase in debris cover is not restricted to the few small glaciers like Schnapfenkuchl H and V. It is a widespread phenomenon in the Austrian Silvretta. For example, length change measurements of Litzner glacier were abandoned because the identification of the glacier margin was hampered by the increase in debris cover. Figure 5 illustrates that this process took

only about a decade, with bare ice clearly evident in the orthophoto of 2006, but not in 2012 and 2015. Moreover, the higher glaciers, like Verhupf glacier, are affected in a way which potentially impacts on the accuracy of glacier delineation: With



melt rates of more than 1 m w.e. even close to the summits, the flat glacier disintegrates, as measured on Jamtalferner (Fischer et al., 2016c). In the course of disintegration and surface elevation lowering, debris and rock fall cover the ice surface.

We will very likely face a rapid recession of mountain glaciers in the coming decades, with large glacier systems disintegrating into smaller glaciers. This has already been happening in the Silvretta for the last hundred years. The now very small and rapidly changing glaciers of the Austrian Silvretta are therefore a perfect test site for analysing the potentials and limitations of repeat LiDAR as a high-resolution airborne remote-sensing method for monitoring glacier fade out in qualitative and quantitative terms. Even small glaciers contribute to sea level rise (Bahr and Radic, 2012) and can be significant for local

hydrological and hazard management. The precise monitoring of a glacial landscape evolving into a postglacial one is also important for the interpretation and dating of paleoglacial landforms, as studied in the Austrian Silvretta by Braumann et al., 2020.

By definition, the new glacier inventory aims at tackling the changes in area and volume of all glaciers in the region. This raises the research questions which of the potentially transient cryogenic structures should remain part of a glacier inventory,

and what the effect of neglecting ice remnants on inventory data would be.

To answer the research questions related to the compilation of glaciers inventories in a stage of early deglaciation, this study presents

|       |     |                                                                                                          |
|-------|-----|----------------------------------------------------------------------------------------------------------|
| I)    |     | a new glacier inventory based on two high-resolution LiDAR DEMs to identify and quantify glacier areas in |

125                2017/18,

II)    the rate of area and volume changes since the last glacier inventories of 1850, 1969, 2002 and 2004/2006

III)    a discussion of the potential and limitations of repeat LiDAR to map glacier changes under current glacier states

IV)    a discussion of glacier inventory strategies to monitor transient glacier states under conditions of beginning deglaciation.


## 2    Data and Methods

### 2.1    Orthophotos 2015/2018 and surface elevations for 2017/2018

Surface elevation data for 2017/18 were made available by the federal administrations of the provinces of Tyrol and Vorarlberg (see the locations of the glaciers in Figure 1). The orthophotos of Tyrol date from summer 2015 for the southern part and from



2018 for the northern part (data.tirol.gv.at, 2020). The orthophotos of Vorarlberg date from 2018 and are available via the WMS Service geoland.at (2020). Spatial resolution of the images is 0.2-0.5 m.

The airborne LiDAR Digital Elevation Models (DEMs) used for mapping glaciers in this study date from 2017 (Vorarlberg) and 2018 (Tyrol), with horizontal resolutions of 0.5x0.5 m (Vorarlberg) and 1x1 m (Tyrol).

The LiDAR DEMs of 2017 and 2018 were both coregistered to the national GIS Grid (BEV, 2011a) with elevation (BEV, 140 2011b) as a national standard procedure. The coregistration of the high-resolution full waveform LiDAR DEMs to the earlier LiDAR DEMs is not considered state of the art of LiDAR technology (Attwenger, personal communication).
For the LiDAR data of Vorarlberg, the Silvretta was covered by 76 flight stripes during 14 and 29 August 2017. Quality control was carried out with ORIENT-LIDAR, resulting in a σ0 of 0.028 m, a RMS for pass area residuals of 0.004 / 0.004 / 0.016 m in x/y/z and 0.056 / 0.056 / 0.031 m for all points (Würländer, 2019). In the final report for the Tyrolean part (Rieger,2019) 145 the uncertainty estimate of the LiDAR data processed with the OPALS software is estimated by the comparison to control areas. The resulting standard deviation of elevation at the control areas is 0.032 m. The control area located in the subsample we used for the study showed a standard deviation of 0.030 m. All control areas are located on stable ground outside glaciers.

## 2.2 Areas and surface elevations in previous glacier inventories

For the Austrian part of the Silvretta range, we compiled glacier area (A) changes to the LIA maximum from mapping moraines, for 1969, 1996 and 2002 from orthophotos, and for 2004 (Vorarlberg) and 2006 (Tyrol) from existing glacier inventories (Fischer et al., 2015). For all inventories apart from the LIA inventory, not only glacier areas but also glacier surface elevations are available. The glacier margins were delineated manually with an uncertainty of the resulting area (σA) of ± 1.5**%** for glaciers larger than 1km² and ±5**%** for smaller ones (Abermann et al. 2009).

The uncertainty of surface elevations on glaciers as quantified by Abermann et al. (2010) is less ±0.3 m for the DEMs based on the LiDAR flights in 2004 and 2006. The federal administration of Tyrol quantifies the single point standard deviation as 0.03 m based on the analysis of pass areas (Federal Government of Tyrol, 2020). The federal administration of Vorarlberg evaluated their DEM of 2004 with 58.597 object elevations and 49.483 terrestrially measured points of different types and found a mean elevation difference of -0.04 m and -0.05 m respectively. 95% of the points meet an accuracy of ±0.4 m 160 (Landesvermessungsamt Feldkirch, 2004).

## 2.3     Compilation of the glacier inventory 2017/2018



The glacier outlines were mapped manually based on the high-resolution LiDAR shaded reliefs and volume changes generated

from LiDAR DEMs (Figure 6, Abermann et al., 2009) with 1x1 m pixel size. The shaded reliefs allow a first estimate of the

outline by distinguishing the smooth glacier areas from the rougher peri- and paraglacial area. The characteristic patterns of

elevation changes reaching their maximum at the glacier terminus help to include debris-covered glacier areas as well.

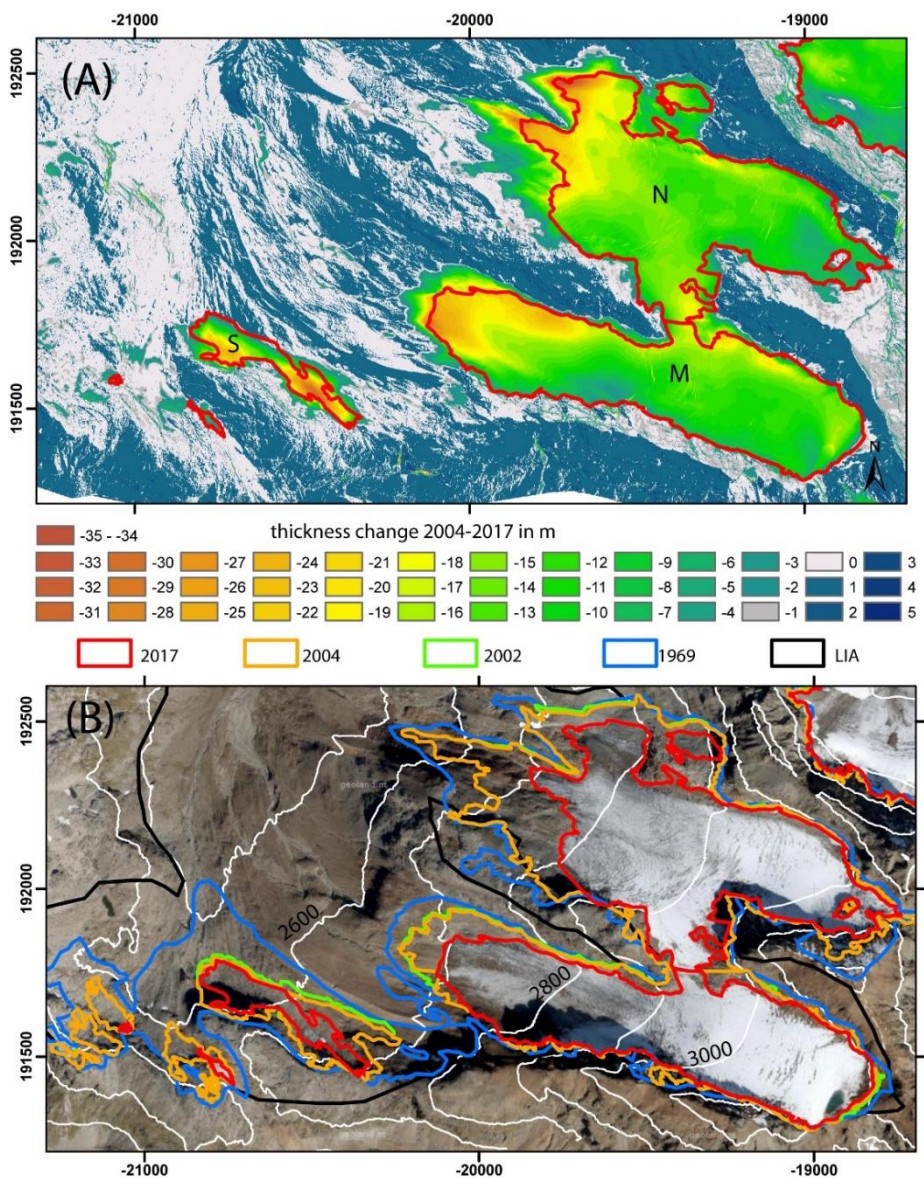

**Figure 6: For the new glacier inventory of the Austrian Silvretta, glacier boundaries were mapped manually from**

**thickness changes between 2004-2017 (A), with the glacier margin positioned at the maximum thickness change**





**(Abermann et al., 2009). Resulting glacier boundaries are displayed on the orthophoto of 2018 for Klostertaler Gletscher (B). Source orthophoto: CC 4.0 https://www.data.gv.at/katalog/en/dataset/orthofoto.**

The identification of areas with subsidence by melt is a clear advantage of this technology and allows mapping glacier areas
that are hard to identify in orthophotos or VIS remote-sensing data. In a validation step, orthophotos were used to check the LiDAR-derived outlines.

We applied no minimum size to mapping the glaciers. Only four of the 46 glaciers mapped were larger than 1 km² in 2002, and two of the glaciers were smaller than the 0.01 km² recommended by Paul et al. (2009) as a practical lower limit for mapping mountain glaciers by remote sensing.


2.4      **Comparison of nominal relative uncertainties of all LiDAR data used in the study**

LiDAR surveys using different instruments and intended point densities (Table 1) produce different representations of the infinitesimally accurate 'real surface', even without real surface changes over time. Based on achieved point densities, the
gridding method and resolution may add misalignment of different DEM of the same area. For this, LiDAR data is validated at defined reference areas. Nevertheless, there is a tradition in glaciological remote sensing and photogrammetry to cross-check the DEM accuracy for glacier-covered areas in potentially stable areas without surface changes.

**Table 1: Instrument, minimum point density and pixel size of the LiDAR campaigns used in this study for the**
**calculation of volume changes.**

| Federal region | Year of survey | LiDAR instrument | Minimum point density per m² | DEM cell size |
|---|---|---|---|---|
| | | | | |
| | | | | m |
| Tyrol | 2006 | ALTM 3100 and Gemini | 0.25 | 1.0 |
| Tyrol | 2018 | Riegl VQ 780 | 8.00 | 1.0 |
| Vorarlberg | 2004 | ALTM 2050 | 2.50 | 1.0 |
| Vorarlberg | 2017 | Riegl LMS-Q780 | 6.20 | 0.5 |



To estimate the final uncertainty with respect to changing point densities and methods applied for generating DEMs (Table 2), we analysed surface elevation changes (Δz) not only at glaciers, but also at rock glaciers and for a buffer of 1000 m and between 1000 to-2000 m around all glaciers and rock glaciers. Although these areas are only partly representative for glaciers

in terms of slope (Figure 7) and roughness, we consider these numbers a very conservative estimate for the uncertainty of the Δz at glaciers. For the rough and changing rock glaciers, we found a mean Δz of -0.4 m with a standard deviation of 1.1 m. At the buffer, excluding the unstable paraglacial areas (1000-2000m), we found a mean elevation difference of 0.0±0.6 m. The standard deviation is twice the uncertainty found by Abermann for LiDAR DEMs. We took the standard deviation as error in Δz for further error propagation.


**Table 2: Comparison of the slope and elevation change (mean, standard deviation) between 2004/06 and 2017/18 for 3 subsamples.**

|  |  | Slope |  | Elevation change (Δz) |  |
|---|---|---|---|---|---|
|  | area | mean | σ | mean | σ |
| Subsamples | km² | ° | ° | m | m |
| rock glaciers | 7.35 | 26 | 10 | -0.4 | 1.1 |
| buffer 1000 m | 112.85 | 33 | 15 | -0.1 | 0.9 |
| buffer 1000 - 2000 m | 28.92 | 27 | 15 | 0.0 | 0.6 |

Studies on the derivation of DEMs from LiDAR point clouds reveal that a slope steeper than about 40° potentially exhibits

larger deviation from the 'true' surface (Sailer et al., 2014). Although the algorithm applied to convert point clouds to gridded data plays a major role for the representation of a specific surface (elevation and shape), the representation of the smooth glacier surfaces is a bit more resilient to low resolution than very rough geomorphological features. Sailer et al. (2014) claim that the cell size for analysing glacier changes could be even between 5 and 10 m.

Sailer et al. (2014) recommend cell sizes below 1 m for terrain steeper than 40°, which we rarely find on the glaciers of the

Austrian Silvretta. There, 90% of the glacier area presents slopes below 40° (Figure 7). In any case, the spatial resolution of the LiDAR DEMs analysed in this study fulfils the criteria above.





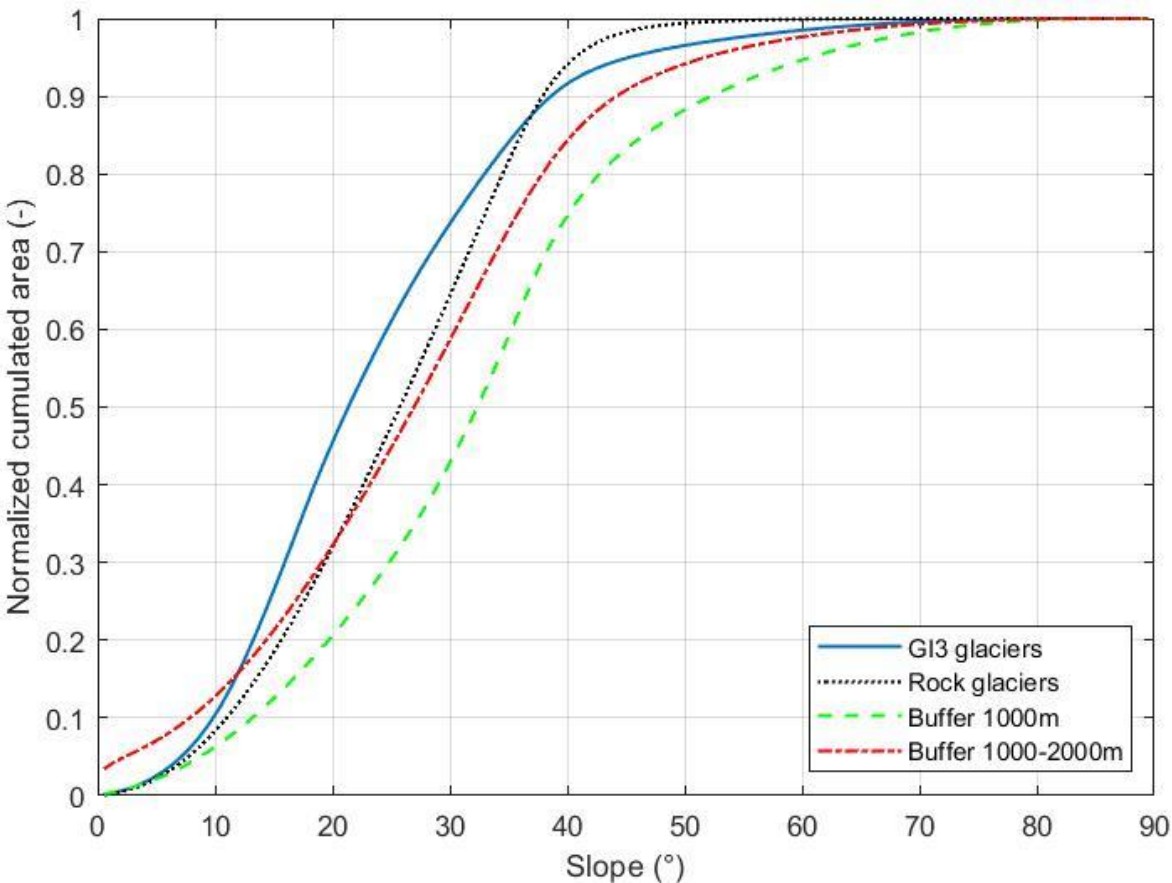

**Figure 7: Distribution of slopes for glaciers, rock glaciers and the two buffer areas used for validation. The buffer regions are steeper than the glaciers, so that the elevation difference in the stable buffer zones can be considered an upper limit for the uncertainty.**

**2.4     Volume change**

The volume change for each pixel was calculated by multiplying the $\Delta z$ with the 1x1 m² pixel area A. For the total volume change $\Delta V$ of a glacier, the volume changes of all pixels within the glacier margins of the first date (t1) of the period t1 to t2 (Equation 1) were summed up.

$\Delta V = \sum A * \Delta z$     for all pixels part of the glacier area at $t_1$        (1)


The maximum uncertainty of Δz at one pixel is the sum of the uncertainties in the elevations of both DEMs involved. The uncertainty in area adds to the uncertainties in Δz for the uncertainty of the total volume change.

Presuming that Δz and ΔA are independent variables, the uncertainties in area and thickness change propagate to the volume change uncertainty (Equation 2).

$$\sigma_{\Delta V} = |\Delta V| \sqrt{\left(\frac{\sigma A}{A}\right)^2 + \left(\frac{\sigma \Delta z}{\Delta z}\right)^2}$$
(2)

**2.5 Geodetic mass balance**

The geodetic mass balance $B_{geo}$ was calculated from volume change, assuming a constant glacier density $\rho$ of 850 kg/m³ (Equation 3).

$$B_{geo} = \Delta V * \rho$$
(3)

Calculating the geodetic mass balance from volume change requires assumptions on the stability of the glacier bed and on the density of the volume lost or gained. Erosion and deposition of sediments at the glacier bed was neglected for this study, as research on the quantification of volumes is still ongoing. Previous studies on Austria's mass balance glaciers used a constant density of 850 kg/m³, so did Fischer et al. (2015) for the Swiss glaciers.

In recent years, the firn cover has melted. Thus the density at the surface may be higher. At the same time, ice flow velocities
have slowed down (Stocker-Waldhuber et al., 2019). This has resulted in the formation of englacial cavities (Stocker-Waldhuber et al., 2017), which reduces the average density at the glacier tongues. The spatial and temporal variability of glacier density σρ is expressed in an uncertainty of ±60 kg/m³. Rocks, debris and sediments on and within the glacier are treated as being part of the glacier.

$$\sigma_{Bgeo} = |B_{geo}| \sqrt{\left(\frac{\sigma \rho}{\rho}\right)^2 + \left(\frac{\sigma \Delta V}{\Delta V}\right)^2}$$
(4)

The annual area-averaged specific geodetic mass balance $b_{geo}$ is then calculated by Equation 5, dividing the geodetic mass balance $B_{geo}$ by the area of the glacier at the beginning of period $t_1$ and by the number of years $(t_2-t_1)$.

$$b_{geo} = \frac{B_{geo}}{A(t_1)*(t_2-t_1)}$$
(5)

The uncertainty of the annual area-averaged specific mass balance is then defined as (Equation 6)

$$\sigma_{bgeo} = |b_{geo}| \sqrt{\left(\frac{\sigma Bgeo}{Bgeo}\right)^2 + \left(\frac{\sigma A(t1)}{A(t1)}\right)^2}$$
(6)



## 3 Results

### 3.1 Total area and volume changes


From the LIA maximum to 2017/18, the Austrian Silvretta lost 68% of its glacier area (Table 3). The mean annual area loss in the latest period was -2.4%, which is more than twice the loss of the period before. The 10 totally debris-covered glaciers cover

an area 0.303 km² (Table S3). For three glaciers (ID 13006, Fluchthornferner S and Litzner Gletscher E), neither bare ice nor signs of motion or a drainage system were visible, mean thickness changes between 2004/06 and 2017/18 were smaller than 2.6 m without a clear thickness change pattern to indicate an ice margin. From that we can conclude that the subsurface ice merely melted.

Area changes were calculated for all glaciers for the periods LIA maximum -1969 and 1969-2002 (Table 3). The dates of

LiDAR campaigns differ between the federal states of Vorarlberg (2004 and 2017) and Tyrol (2006 and 2018). Later periods are thus 2002-2004/06, and 2004/06-2017/18.

**Table 3: Glacier area in the Austrian part of the Silvretta between LIA maximum, 1969, 2002, 2004/06 and 2017/18. For Vorarlberg, LiDAR flights were carried out in 2004 and 2017, for Tyrol in 2006 and 2018.**

| Time | Area | | % of LIA area | | Period | Area change | | | | | |
|---|---|---|---|---|---|---|---|---|---|---|---|
| | | km² | | | | km² | | % | | % per yr | |
| LIA maximum | 40.9 ± | 4.1 | 100 ± | 10 | | | | | | | |
| 1969 | 24.0 ± | 0.8 | 59 ± | 2 | LIA-1969 | 17.0 ± | 4.9 | -41 ± | 12 | | |
| 2002 | 19.1 ± | 0.6 | 47 ± | 3 | 1969-2002 | -4.8 ± | 1.4 | -20 ± | 6 | -0.6 ± | 0.2 |
| 2004/06 | 18.5 ± | 0.3 | 45 ± | 1 | 2002-2004/06 | -0.6 ± | 0.9 | -3 ± | 5 | -1.1 ± | 1.6 |
| 2017/18 | 13.1 ± | 0.4 | 32 ± | 2 | 2004/06-2017/18 | -5.4 ± | 0.7 | -29 ± | 4 | -2.4 ± | 0.3 |


The annual specific geodetic mass balance (Table 4) was highest in the short period with the extreme summer of 2003 (-1.5±0.7 m w.e./y), and lowest in the period 1969-2002 (-0.2±0.1 m w.e). That geodetic mass balance in the latest period is lower than in the short period before, despite extreme losses evident in the mass balance time series (Fischer et al., 2016b) can

be attributed to the loss of ablation area within the period, reducing the overall volume loss as no ice to melt is left in the areas



with highest ablations in the past. This is also evident from the fact that the maximum specific direct mass balance has been measured in 2015 (Fischer et al., 2016), despite an increase of mass loss at most of the (remaining) stakes.

**Table 4: Glacier volume loss, geodetic balance (total, specific and specific annual) in the Austrian part of the Silvretta**
**between 1969, 2002, 2004/06 and 2017/18. For Vorarlberg, LiDAR flights were carried out in 2004 and 2017, for Tyrol in 2006 and 2018.**

|  | Volume loss | Geodetic balance | Specific geodetic balance | Annual specific geodetic balance |
|---|---|---|---|---|
| Period | km³ | km³ w.e. | m w.e. | m w.e./yr |
| 1969-2002 | -0.213 ± 0.092 | -0.181 ± 0.079 | -7.6 ± 3.3 | -0.2 ± 0.1 |
| 2002-2004/06 | -0.118 ± 0.048 | -0.100 ± 0.042 | -5.2 ± 2.2 | -1.5 ± 0.7 |
| 2004/06-2017/2018 | -0.237 ± 0.024 | -0.201 ± 0.025 | -10.9 ± 1.4 | -0.8 ± 0.1 |

**3.2 Volume changes and geodetic balance for individual glaciers**

For individual glaciers, the three inventories also show strongest annual losses for the short period with the extreme year 2003
(Figure 8, Table 5), lowest annual losses for all glacier sizes in the period 1969-2002 and medium losses for the latest period. Glaciers with smaller areas present the highest variability. In contrast to the short and first warm inventory period 2002-2004/06, when some of the smaller glaciers were still quite stable, the range of geodetic mass balance for small glacier sizes is extremely high, from losses of more than 1 m w.e./ year to a few quite stable conditions, which are related to debris cover (or loss of ice).



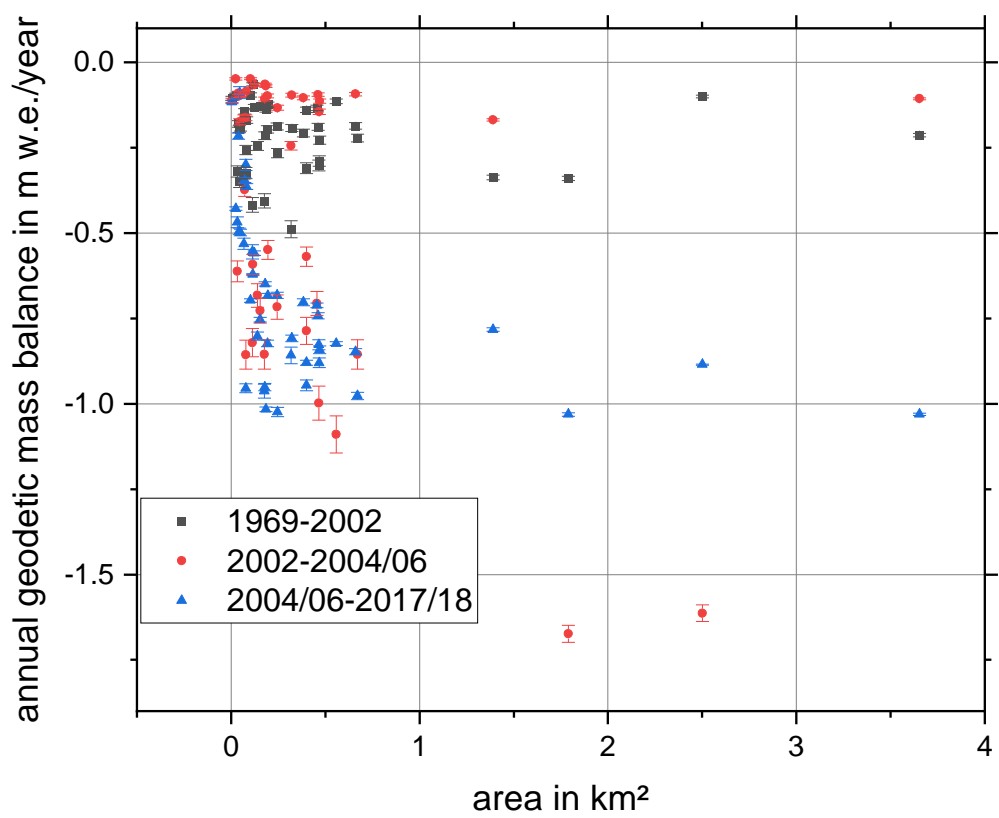

**Figure 8: Annual geodetic mass balances for the Austrian Silvretta in 1969-2002, 2002-2004/06, 2004/06-2017/18.**


**Table 5: Glaciers of different classes (c), areas for the year (y) of the LiDAR survey, geodetic balance (total, specific and specific annual) in the Austrian part of the Silvretta between 2004/06 and 2017/18. For Vorarlberg, LiDAR flights**
**were carried out in 2004 and 2017, for Tyrol in 2006 and 2018. Classes: d… widely debris covered, g…gone. For glaciers 13004 and 13005, no elevation data were available, so that the area was mapped with orthophotos.**

| name | c | ID (AT) | y | area | | y | area | | volume change | annual geodetic balance |
|---|---|---|---|---|---|---|---|---|---|---|
| | | | | km² | | | km² | | $10^6 m^3$ | m w.e./yr |
| Jamtal F. | | 13019 | 06 | 3.65 ± | 0.05 | 18 | 2.792 ± | 0.042 | -53.2 ± 4.46 | -1.0 ± 0.2 |
| Ochsentaler G. | | 12008 | 04 | 2.50 ± | 0.04 | 17 | 2.135 ± | 0.032 | -33.9 ± 3.05 | -0.9 ± 0.2 |



| | | | | | ± | | | | ± | | | ± | | | ± | |
|---|---|---|---|---|---|---|---|---|---|---|---|---|---|---|---|---|
| Vermunt G. | | 12007 | 04 | 1.79 | ± | 0.03 | 17 | 1.234 | ± | 0.019 | -28.2 | ± | 2.19 | -1.0 | ± | 0.2 |
| Larain F. | | 13007 | 06 | 1.39 | ± | 0.02 | 18 | 0.989 | ± | 0.049 | -15.4 | ± | 1.68 | -0.8 | ± | 0.2 |
| Schneeglocken G. | | 12009 | 04 | 0.67 | ± | 0.03 | 17 | 0.539 | ± | 0.027 | -10.0 | ± | 0.95 | -1.0 | ± | 0.2 |
| Totenfeld | | 13021 | 06 | 0.66 | ± | 0.03 | 18 | 0.523 | ± | 0.026 | -7.90 | ± | 0.89 | -0.8 | ± | 0.2 |
| Klostertaler G. N | | 12013 | 04 | 0.56 | ± | 0.03 | 17 | 0.369 | ± | 0.018 | -7.04 | ± | 0.76 | -0.8 | ± | 0.2 |
| Futschöl F. | | 13014 | 06 | 0.47 | ± | 0.02 | 18 | 0.368 | ± | 0.018 | -5.58 | ± | 0.63 | -0.8 | ± | 0.2 |
| Bieltal F. | | 13028 | 06 | 0.47 | ± | 0.02 | 18 | 0.280 | ± | 0.014 | -5.81 | ± | 0.63 | -0.9 | ± | 0.2 |
| Schattenspitz G. | | 12011 | 04 | 0.46 | ± | 0.02 | 17 | 0.373 | ± | 0.019 | -5.88 | ± | 0.63 | -0.8 | ± | 0.2 |
| Chalaus F. | | 13017 | 06 | 0.46 | ± | 0.02 | 18 | 0.358 | ± | 0.018 | -4.83 | ± | 0.60 | -0.7 | ± | 0.2 |
| Litzner G. | | 12021 | 04 | 0.46 | ± | 0.02 | 17 | 0.228 | ± | 0.011 | -4.96 | ± | 0.60 | -0.7 | ± | 0.2 |
| Klostertaler G. M | | 12014 | 04 | 0.40 | ± | 0.02 | 17 | 0.334 | ± | 0.017 | -5.39 | ± | 0.55 | -0.9 | ± | 0.2 |
| Rauhkopf G. | | 12005 | 04 | 0.40 | ± | 0.02 | 17 | 0.257 | ± | 0.013 | -5.79 | ± | 0.56 | -0.9 | ± | 0.2 |
| Fluchthorn F. | | 13011 | 06 | 0.38 | ± | 0.02 | 18 | 0.232 | ± | 0.012 | -3.81 | ± | 0.50 | -0.7 | ± | 0.2 |
| Mittlere Schnapfenkuchl | | 13009 | 06 | 0.32 | ± | 0.02 | 18 | 0.242 | ± | 0.012 | -3.68 | ± | 0.43 | -0.8 | ± | 0.2 |
| Kronen F. | | 13013 | 06 | 0.32 | ± | 0.02 | 18 | 0.146 | ± | 0.007 | -3.86 | ± | 0.43 | -0.9 | ± | 0.2 |
| Jamtal F. W | | 13020 | 06 | 0.25 | ± | 0.01 | 18 | 0.184 | ± | 0.009 | -3.55 | ± | 0.34 | -1.0 | ± | 0.2 |
| Im Glötter S | | 12016 | 04 | 0.25 | ± | 0.01 | 17 | 0.108 | ± | 0.005 | -2.56 | ± | 0.32 | -0.7 | ± | 0.1 |
| Verhupf G. | | 12019 | 04 | 0.19 | ± | 0.01 | 17 | 0.073 | ± | 0.004 | -2.03 | ± | 0.25 | -0.7 | ± | 0.1 |
| Rosstal F. | | 13024 | 06 | 0.19 | ± | 0.01 | 18 | 0.131 | ± | 0.007 | -2.24 | ± | 0.26 | -0.8 | ± | 0.2 |
| Getschner F. | | 13023 | 06 | 0.19 | ± | 0.01 | 18 | 0.154 | ± | 0.008 | -2.66 | ± | 0.26 | -1.0 | ± | 0.2 |
| Madlener F. | | 13026 | 06 | 0.18 | ± | 0.01 | 18 | 0.125 | ± | 0.006 | -1.65 | ± | 0.23 | -0.6 | ± | 0.1 |
| Henneberg F. | | 13025 | 06 | 0.18 | ± | 0.01 | 18 | 0.113 | ± | 0.006 | -2.40 | ± | 0.25 | -1.0 | ± | 0.2 |
| Tiroler G. | | 12006 | 04 | 0.18 | ± | 0.01 | 17 | 0.101 | ± | 0.005 | -2.61 | ± | 0.25 | -1.0 | ± | 0.2 |
| Schweizer G. | | 12023 | 04 | 0.15 | ± | 0.01 | 17 | 0.068 | ± | 0.003 | -1.76 | ± | 0.20 | -0.8 | ± | 0.2 |
| NN | d | 12012 | 04 | 0.14 | ± | 0.01 | 17 | 0.091 | ± | 0.005 | -1.71 | ± | 0.19 | -0.8 | ± | 0.2 |
| NN | | 13022 | 06 | 0.12 | ± | 0.01 | 18 | 0.083 | ± | 0.004 | -0.97 | ± | 0.16 | -0.6 | ± | 0.1 |
| Kromer G. | | 12022 | 04 | 0.11 | ± | 0.01 | 17 | 0.062 | ± | 0.003 | -1.09 | ± | 0.15 | -0.6 | ± | 0.1 |
| Klostertaler G. S | d | 12015 | 04 | 0.11 | ± | 0.01 | 17 | 0.040 | ± | 0.002 | -0.95 | ± | 0.14 | -0.6 | ± | 0.1 |
| Oberer Augsten F. | | 13016 | 06 | 0.10 | ± | 0.01 | 18 | 0.089 | ± | 0.004 | -1.02 | ± | 0.13 | -0.7 | ± | 0.2 |
| Hintere Schnapfenkuchl | d | 13008 | 06 | 0.082 | ± | 0.004 | 18 | 0.025 | ± | 0.001 | -0.42 | ± | 0.10 | -0.4 | ± | 0.1 |
| Unterer Augsten F. | d | 13015 | 06 | 0.078 | ± | 0.004 | 18 | 0.036 | ± | 0.002 | -0.33 | ± | 0.10 | -0.3 | ± | 0.1 |



| | | | | | | | | | | | | | | | | | |
|---|---|---|---|---|---|---|---|---|---|---|---|---|---|---|---|---|---|
| Im Glötter N | | 12017 | 04 | 0.078 | ± | 0.004 | 17 | 0.049 | ± | 0.002 | -1.14 | ± | 0.11 | -1.0 | ± | 0.2 |
| NN | d | 12018 | 04 | 0.073 | ± | 0.004 | 17 | 0.021 | ± | 0.001 | -0.38 | ± | 0.09 | -0.3 | ± | 0.1 |
| Bieltal F. E | | 13027 | 06 | 0.068 | ± | 0.003 | 18 | 0.019 | ± | 0.001 | -0.51 | ± | 0.09 | -0.5 | ± | 0.1 |
| NN | | 13004 | 06 | 0.064 | ± | 0.003 | 18 | 0.043 | ± | 0.002 | | | | | | |
| NN | d | 13005 | 06 | 0.052 | ± | 0.003 | 18 | 0.033 | ± | 0.002 | | | | | | |
| NN | | 13018 | 06 | 0.050 | ± | 0.003 | 18 | 0.030 | ± | 0.001 | -0.35 | ± | 0.06 | -0.5 | ± | 0.1 |
| Fluchthorn F. S | g | 13012 | 06 | 0.046 | ± | 0.002 | 18 | 0.000 | ± | 0.000 | -0.06 | ± | 0.05 | -0.1 | ± | 0.1 |
| Vordere Schnapfenkuchl | d | 13010 | 06 | 0.043 | ± | 0.002 | 18 | 0.028 | ± | 0.001 | -0.30 | ± | 0.05 | -0.5 | ± | 0.1 |
| NN | g | 13006 | 06 | 0.039 | ± | 0.002 | 18 | 0.000 | ± | 0.000 | -0.12 | ± | 0.05 | -0.2 | ± | 0.1 |
| Schattenspitz G. E | d | 12010 | 04 | 0.033 | ± | 0.002 | 17 | 0.006 | ± | 0.000 | -0.24 | ± | 0.04 | -0.5 | ± | 0.1 |
| Garnera G. | d | 12026 | 04 | 0.030 | ± | 0.002 | 17 | 0.011 | ± | 0.001 | -0.14 | ± | 0.04 | -0.3 | ± | 0.1 |
| Platten G. | d | 12025 | 04 | 0.026 | ± | 0.001 | 17 | 0.012 | ± | 0.001 | -0.17 | ± | 0.03 | -0.4 | ± | 0.1 |
| Litzner G. E | g | 12020 | 04 | 0.0038 | ± | 0.0002 | 17 | 0.000 | ± | 0.001 | -0.01 | ± | 0.00 | -0.1 | ± | 0.1 |

## 4 Discussion

### 4.1 Challenges for mapping area changes of disintegrating glaciers

With ongoing glacier disintegration, mapping glacier outlines becomes ever more ambiguous even if using high-resolution volume change data. Major points to discuss are:

- what exactly are the properties that make a cryogenic feature a glacier which should be included in a glacier inventory,
- should we introduce inventories of all cryogenic features including glaciers, permafrost and rock glaciers or
- do we need to define a point of glacier disappearance?

All glaciers in this study were mapped first in previous inventories at times when bare ice was largely visible. For the now totally debris-covered glaciers, it is hard to decide using only optical data if there is any ice left. In the absence of bare ice or stable surface structures, such as ponds or crevasses, and with soft slopes and low potential velocities, surface velocities do not help to distinguish buried glaciers from rock glaciers and permafrost dynamics. As measurements in bore holes in rock glaciers show, sliding debris and rocks on the ice can account for a major part of the total surface velocity (Krainer et al., 2015). If a subsurface runoff system exists at the terminus, it is necessary to analyse if it is fed by groundwater, melt of seasonal snow, permafrost ice or glacier ice. This can be done by analysing seasonality of the amount of runoff (e.g. Brighenti



et al., 2019), chemical composition and ecological properties (e.g. Tolotti et al., 2020) as well as by isotope analysis of the meltwater (e.g. Wagenbach et al., 2012).

Careful analysis is needed to decide whether a formerly well-defined glacier still fulfils the criteria of a glacier. Taking glaciers off inventories prematurely is to be avoided, as they may still contribute to glacial runoff in the basin, can force debris slide or

serve as essential climate variables. Without keeping them in inventories, we lose track of these transient states.

## 4.4 Uncertainties in mapping totally debris-covered glaciers

Mapping glaciers that have become fully covered with debris is uncertain for two reasons. First, mapping areas of possible ice

covered with debris by volume changes only allows tackling the presence of melting ice during the inventory period, but it does not prove that any ice is left at the end of the period. Second, the presence of melting ice is not restricted to debris-covered glaciers but is also true for permafrost. The Schnapfenkuchl glaciers V and H (Figure 9) present glacier ice at locations where debris flows exposed the ice and past inventories showed bare ice (see the supplement).

The Schnapfenkuchl glaciers are embedded in an environment adjacent to a number of rock glaciers. The delineation of buried

glaciers in the presence of permafrost and mass movements upon and from the glacier needs a high temporal frequency of inventory data to arrive at the detection of glacier ice. High spatial resolution is needed to distinguish between ice loss and ice dynamics and to track the geomorphological processes and features related to volume change.


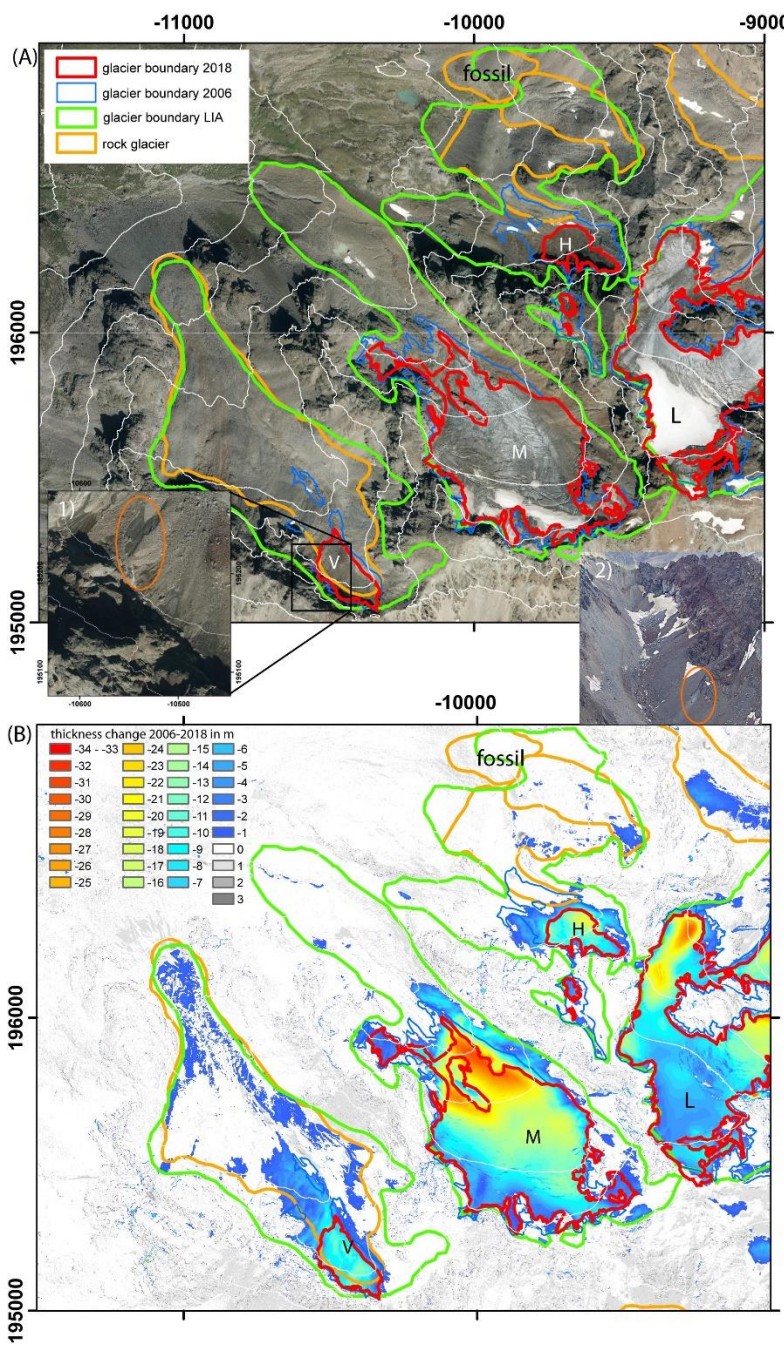

**Figure 9: Schnapfenkuchl glaciers V, M and H with their 2018, 2006 and LIA extent displayed on (A) the orthophoto from 2015 and (B) the elevation changes from 2006-2018. Inserts show debris flows exposing bare ice on the orthophoto (1) and an airborne photo (2) dating from 20.08.2020 with bare ice exposed in another location. Source orthophoto: CC 4.0 https://www.data.gv.at/katalog/en/dataset/orthofoto. The only fossil rock glacier is indicated in (A) and (B), all other rock glaciers are classified as active. Find the enlarged versions of the aerial photographs (1), (2) in Fig. 4 and in the supplement.**



## 4.5 Distinguishing rock glaciers from small totally debris-covered glaciers

Past definitions of glaciers focussed on the mass of ice. From the earliest definitions, e.g. by Walcher (1773) and Tyndall (1860), to more modern ones within glaciology (Klebelsberg, 1948) and in neighbouring fields (Dexter, 2013), scientists have obviously been investigating glaciers closer to equilibrium than those in our study:

*"Diese Eisberge werden Ferner genennet, welches Wort… das Eis bedeutet welches mit Schnee vermenget gesammelt hat"* (Walcher, 1773) [These mountains of ice are called „Ferner", a word …that denotes ice mixed with snow which has

accumulated]

*"At its origin then a glacier is snow — at its lower extremity it is ice."* (Tyndall, 1860)

***Gletscher sind Massen körnigen Firns und Eises***, *die aus Schneeansammlungen hervorgehen und sich dahin bewegen, wo sie abschmelzen oder verdunsten können….scheinbarer oder auch wirklicher Mangel an Bewegung schliesst aber doch die Bezeichnung Gletscher nicht aus, übergeordnetes Merkmal ist die Körnerstruktur." (Klebelsberg 1948)*

*[Glaciers are masses of granulated firn and ice that have evolved out of accumulations of snow and which move towards where they can melt or evaporate… however, an apparent or actual lack of movement does not preclude it being labelled a glacier, the predominant characteristic is the granular structure.]*

*"A glacier is a mass of relatively slow moving ice created by the accumulation of snow"* (Dexter, 2013)

Although englacial and supraglacial debris was present on and in glaciers even during the LIA, ice and snow dominated at the

time. In our glacier inventory of 2017/18, 10 of 43 glaciers are predominantly covered by debris, not snow, with only few remnants of bare ice visible. Mean thickness changes range from -1.5±0.1 m to -12.3±0.6 m for the total period. At Garnera Gletscher and glaciers #12018, #13005, no bare ice is visible. On the surface of all other glaciers, bare ice is visible in parts. Several of the debris-covered glaciers, for example, the Schnapfenkuchl glaciers (Figure 9), are located between active rock glaciers captured in the Tyrolean rock glacier inventory (Krainer and Ribis, 2012). This means that two independent groups of

researchers, glaciologists and geologists, mapped the same site, for example, the easternmost Schnapfenkuchl glacier/rock glacier, in inventories of different landforms. A continuum from glacier to debris-covered glacier to rock glacier was recently discussed by Anderson et al. (2018). Kellerer-Pirklbauer and Kaufmann (2018) analysed the glacial history of an Austrian site of long-term rock glacier monitoring and found evidence of post-LIA deglaciation to permafrost formation. For the broader scientific investigation of this phenomenon, international cooperation for comparing processes and regimes in other mountain

regions are vital, since a case study can only be a first step towards deriving more general empirical and theoretical knowledge. More concise definitions of the term *glacier* are needed to reflect current conditions on the ground. We suggest:

1)  Glaciers are bodies of sedimentary ice, firn and snow, formed by densification of snow AND en- and supraglacial sediments of all grain sizes.

or

2)  Glaciers are exposed bodies of sedimentary ice, firn and snow formed by densification of snow with signs of deformation and an en- or supraglacial drainage system.





Definition (1) includes all sizes of ice bodies which have been formed as part of glaciers, even dead debris-covered glacier ice. It understands debris as a natural part of the glacier system, also in the calculation of volume changes, and sets down a clear
base for mapping glaciers in inventories and calculating geodetic balances. It does away with the need for investigating drainage systems or deformation on buried ice bodies to find out if a given structure is a glacier. The drawback is that volume changes are not necessarily ice volume changes, but it accounts for the 'real world' problem that it is not possible to assess the amount of englacial debris anyway.

Definition (2) excludes dead and buried glacier ice, which might have advantages for mapping, especially in low resolutions, but leaves an undefined gap between glaciers and permafrost ice, thereby introducing a third class, i.e. buried glacier ice.

**4.6 Comparison of changes in the Austrian Silvretta with other glacier regions**

For the Austrian Silvretta, the mean annual geodetic balance is -0.8 ±0.1m w.e./yr for the period from 2004/06 to 2017/18.
This is less negative than the -1.03 m w.e. reported for the Glarus and Leopoldine Alps by Sommer et al. (2020) for the period from 2000 to 2014, but but a greater loss than than the -0.62 m w.e./yr calculated by Fischer, Huss and Hoelzle (2015) for the Swiss Alps between 1980 and 2010. They also reported a variability of geodetic balances on the catchment scale, ranging from -0.52 to -1.07 m w.e. Our data confirm the highly variable sensitivity of small glaciers to warming as found by Huss and Fischer (2016).
The annual rate of area losses in the Austrian Silvretta (1969-2017/18) of -1.13% is larger than the -0.52% reported for a slightly shorter period (1986-2014) in the Caucasus (Tielidze et al., 2020), but similar to the -1.1% reported by Paul et al. (2020) for the Alps between 2003 and 2015/16 excluding glaciers smaller than 0.01 km².

**4.7 Additional uncertainties of geodetic mass balance**


Geodetic mass balance has been analysed for potential uncertainties, for example, unaccounted seasonal snow interpreted as ice volume change or changing glacier beds, and real mass changes differing from surface mass balance, for example, as a result of refreezing, firn density changes, or crevasse volume (e.g. Zemp et al., 2013). For the very small and debris-covered glaciers we analysed, an additional source of uncertainty is the amount of the accumulated debris on the surface. This volume
is not part of the hydrological cycle so that, from a hydrological perspective, we would wish to exclude this volume from the analysis. This would be possible only if we could keep track of the erosion rates in the source areas. This would necessitate a totally different monitoring system to track the steep headwalls. In terms of geomorphology, debris and rocks are part of the glacier and therefore there is no need to distinguish old from new supra- and englacial debris and rock.





The rougher surface at steep debris-covered glaciers with avalanche activity seems to encourage the formation of perennial snow patches on the debris-covered glacier. This is not relevant for delineating glaciers but raises the question on the effects of these perennial snow volumes on geodetic mass balance. This question could be also answered with a specific monitoring effort tackling the now stored volumes and water equivalents. In this effort first hints on potential ice formation in such structures could be gathered.

**5 Summary and Conclusions**

The annual change rates of area and volume change indicate an increasing pace of glacier retreat in the Austrian Silvretta. A growing number of nunataks, the disintegration of larger glaciers into smaller ones and the accumulation and relocation of debris on the glacier surface all indicate an additional need for compiling glacier inventories - if we want to keep track of 415 buried glacier ice. Figure 10 outlines the procedure applied in this study. For most of the 46 glaciers, the procedure of delineating the glacier surface by surface roughness and volume change validated by orthophotos worked out well, even for the ten glaciers totally covered by debris. However, for some of the analysed glaciers, for example Schnapfenkuchl V, future analysis of areas will be difficult, as the surface is now totally covered with debris, and accumulation and relocation of the debris is still ongoing. Therefore, we cannot expect a distinct pattern of thickness change in future, with a maximum close to 420 the terminus. In the next decade we thus need a scientific discussion on how to proceed with as yet undefined subsurface glacier remnants.





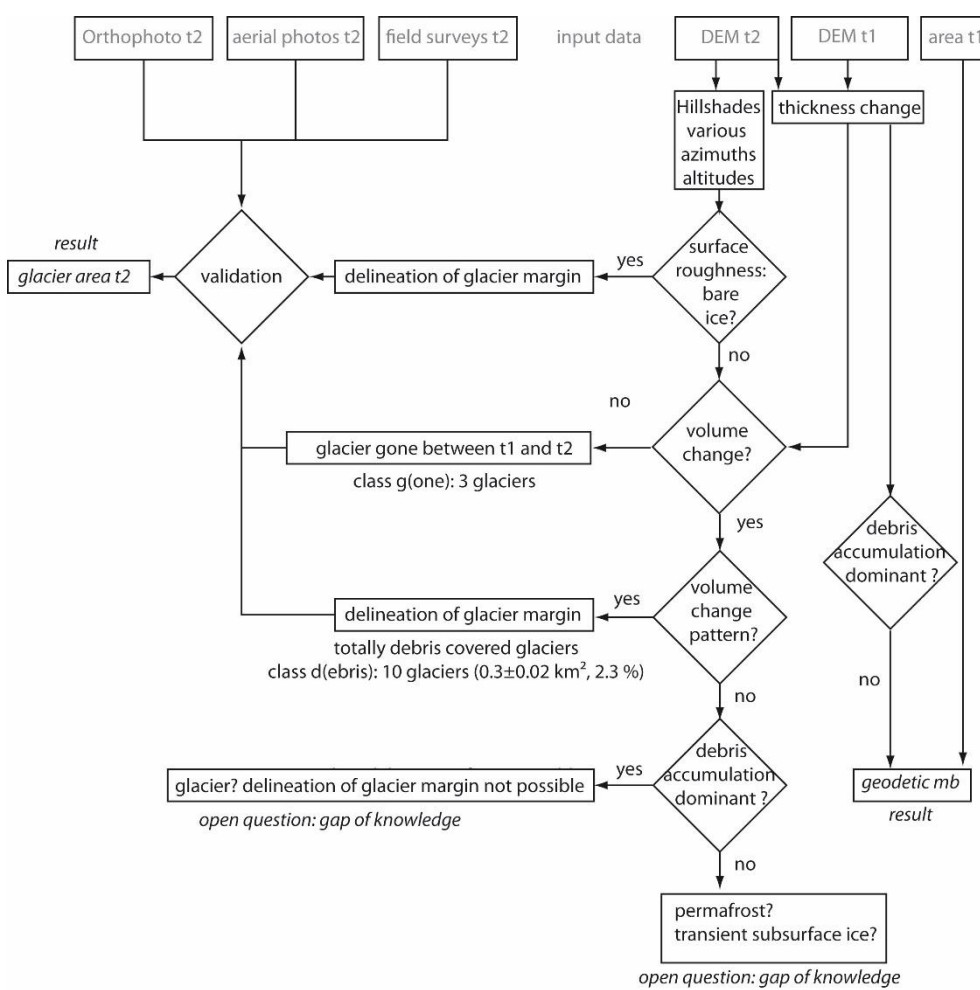

**Figure 10: Workflow for high-resolution glacier mapping and mass balance calculation as applied in this study. It does**
**lead to quantitative results for most of the glaciers but also shows up the need of scientific discussion where**
**geomorphological processes other than ice melt contribute significantly to volume changes.**

The technical requirements for mapping small vanishing glaciers depend on glacier size, annual rate of thickness changes and

accumulation rates of debris. The small glacier structures that remain in the Austrian Silvretta are typically just a few tens of

metres wide. With increasing pixel size, the accuracy of mapping vertical changes decreases, which makes it difficult to

distinguish geomorphological processes like accumulation and relocation of rocks and debris from ice volume change.

Therefore, we recommend 1 m spatial resolution. The vertical accuracy needed to represent the geomorphological processes

of debris relocation depends on the steepness of the area and on the temporal interval chosen. For a period of 10 years and a

slope of up to 40° for Alpine glaciers, a vertical resolution of 1/10 of the spatial resolution is sufficient to distinguish volume

changes by ice melt from erosion and deposition.

Of the now 43 glaciers of the Austrian Silvretta, only three are larger than 1 km², 19 are smaller than 0.1 km² (of these, 13 are smaller than 0.05 km²). Applying minimum sizes for glacier area with thresholds at 0.1 or 0.05 km² would exclude 0.82 km²/6.2% and 0.35 km²/2.7% of the total area. This is higher than /in the same magnitude as the nominal uncertainty of 0.4 km² in the total area. Such thresholds for very small glaciers would make many of them disappear from inventories and

hamper any efforts to tackle the hazard potential of deglaciation, as even small glacier remnants could be relevant here.

There is probably no hard limit for surveying deglaciation in terms of glacier size, as the monitoring strategies for rock glaciers and permafrost can take over. Handing over what is left from glaciers to the scientific networks of the permafrost community could be an emerging and exciting new playground for both fields.

This regional study can only point out the specific challenges and limitations for tackling glacier change in the Austrian

Silvretta. These will differ from region to region depending on the climatic regime, lithology, topography, glacier types and many other factors. We would therefore appreciate an international effort to compare the need for tackling deglaciation in other regions, as a common monitoring framework will be essential in an ever warmer future.

**Author Contributions**

Andrea Fischer designed this study, worked on glaciers in the Silvretta range for over a decade and wrote the text. Kay Helfricht, Bernd Seiser and Martin Stocker-Waldhuber analysed the geodetic data and contributed to discussion and text.

**Funding**

Both LiDAR DEMs were provided by the TIRIS section of the federal administration of Tyrol and the federal administration of Vorarlberg.

**Acknowledgments**

The federal administrations of Tyrol and Vorarlberg are acknowledged for providing geodata. We thank the community of Galtür, the Lorenz family at Jamtalhütte and Oswald Heis for supporting logistics. Kati Heinrich helped with the figures, Brigitte Scott checked the English - thank you!

**Data availability statement**

The glacier inventory data in https://doi.org/10.1594/PANGAEA.844988.



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
