# Peer review of "High-resolution inventory to capture glacier disintegration in the Austrian Silvretta"

_The Cryosphere, 2020_

## Author Comment (AC1)

Response to **RC1**: ['Comment on tc-2020-376'](), Anonymous Referee #1, 21 Feb 2021

RC1

Response 29.03.2021

This is well written and interesting paper that describes glacier disintegration in a region in Austria based on an extensive data set of DEMs and orthophotos. The glaciers in question are all rather small (a few km² or smaller) and some have disintegrated into partly or fully debris-covered patches. The authers point out the need for a scientific discussion about the proper aim and suitable methodology for monitoring small, totally debris-covered cryogenic geomorphological structures that are remnants of recently active glaciers. At some spatial scale and degree of debris-cover, monitoring of such remnants might not be within the proper scope of glacier monitoring as such but should be considered as a task for monitoring and studies of rock glaciers and permafrost. Setting a hard threshold in terms of the size of very small glaciers for including them in glacier inventories and considering then glacier monitoring is, however, not straightforward.

We thank reviewer 1 for their very constructive and helpful comments.

Comments:

p. 14, l. 246: The authors use the area at the first point in time $A(t_1)$ to compute the area-averaged specific mass balance $b_{geo}$. I would have thought that the average of the areas at the beginning and end of the period $(A(t_1)+A(t_2)/2$ would be more suitable? This would be consistent with the description on p. 92 in the IACS glossary of glacier mass balance (Cogley et al., 2011).

We will add b_geo calculated with averaged areas and shift the original version based on the area at the beginning of the period to the supplements.

p. 22, l. 340-353: It might be worthwhile to add the IACS (Cogley et al., 2011, p. 45) and GLIMS (Raup and Khalsa, 2010, p. 4) definitions of glaciers in this list of definitions of the term "glacier" since these have and "official" status in the glaciological community.

We will include:

*"A perennial mass of ice, and possibly firn and snow, originating on the land surface by the recrystallization of snow or other forms of solid precipitation and showing evidence of past or present flow… In contrast to what is natural in dynamic glaciology and glacial geomorphology, for mass-balance purposes the glacier consists only of frozen water. Sediment carried by the glacier is deemed to be outside the glacier."*

Cogley et al., 2011

and the GLIMS definition of glaciers

*"A glacier or perennial snow mass, identified by a single GLIMSglacier ID, consists of a body of ice and snow that is observed at the end of the melt season, or, in the case of tropical glaciers, after transient snow melts. This includes, at a minimum, alltributaries and connected feeders that contribute ice to the main glacier, plus all debris-covered parts of it. Excluded is all exposed ground, including nunataks. .... A stagnant ice mass still in contact with a glacier is part of the glacier, even if it supports an old-growth forest…All debris-covered parts of the glacier must be included…. Rock glaciers and heavily debris-covered glaciers tend to look similar, but their geneses are different. GLIMS does not currently deal with the former, but does include the latter."*

Raup, B. and Khalsa, S.J.S., 2010. GLIMS Analysis Tutorial. GLIMS, Global Land IceMeasurements from Space, NSIDC, www.GLIMS.org
https://www.glims.org/MapsAndDocs/assets/GLIMS_Analysis_Tutorial_a4.pdf

Regarding the definition of "glacier" on p. 22 and 23, it might be mentioned that definition (1), that includes dead and buried glacier ice, would also include ice-cored morains, even with deeply buried ice that may last centuries, which would presumably greatly complicate the creation and management of glacier inventories.

Yes, this is correct. We will add this point to the discussion and add one example of an ice cored moraine.

Minor and editorial comments:

The superscipted power of 2 in "km²" is missing in the pdf I am reading (many places).

We have to check this in the next proof-reading version.

The "-" hyphen in negative numbers and number ranges should be changed to en-dash or a minus sign ("--" in LaTeX), for example "2002-2004" should be "2002--2004"

will be corrected

p. 12, l. 197: "between 1000 to-2000 m" --> "between 1000 to 2000 m"

will be corrected

p. 15, l. 269: "lower than" --> "lower in magnitude than"

will be corrected

p. 19, l. 305: "... if there is any ice left." --> "... whether there is any ice left."

will be corrected

p. 20, l. 315: "Without keeping them in inventories, we lose track of these transient states." --> "We lose track of these transient states by dropping these glaciers from the inventories."
will be corrected

p. 24: "The annual change rates of area and volume change indicate ..." --> "The annual rates of change of area and volume indicate ..."
will be corrected

p. 26: "The glacier inventory data in https://doi.org/10.1594/PANGAEA.844988" --> "The glacier inventory data are available at https://doi.org/10.1594/PANGAEA.844988"

will be corrected

---

## Author Comment (AC2)

Response to Reviewer RC 2

We thank reviewer RC 2 for their very constructive and detailed review which will clearly help to improve this paper by pointing out weaknesses, but also suggesting the respective improvements which we will be necessary to be included for final publication. Please find all the details below.

Review of TC-2020-376 by Fischer et al.

[1] The study by Fischer et al. is presenting for glaciers in the Austrian part of the Silvretta group two different topics: (A) the results of a new glacier inventory and glacier changes (in area and mass) over time and (B) the problems in accurately collecting the required information in times of massive glacier down-wasting and disintegration. Both topics are relevant and of interest, but I miss the connection between them and a more detailed elaboration of cause and effect that goes beyond the presentation of new numbers in tables and some vague speculations about processes that are not presented or observed.

We thank for pointing out that the interconnetion of these two topics is not clear, and that we have to address this more directly than it is done in the first  versionof the manuscript:

The interconnection results from the fact that we  have to address the problems related to the compilations of glacier inventories under the given glacier states before we are able to compile and present a glacier inventory. We suggest to shift the mapping 'problems' to an own section with suggestions, uncertainties and results.

[2] To give two examples here: (1) The difficult identification of glaciers under debris cover is a key aspect of the study (topic B), but only the results of one method (elevation changes from a Lidar dDEM) are shown.

We intended to show that surface elevation change data is essential in addition to optical information for a proper delineation in particular of debris covered glacier parts. Thus, it is not a single method applied, rather an additional method to standard process of delineating glaciers from lidar hillshades, orthophotos or satellite images only.

But following this argument here, we will improve the comparison to other methods of mapping. We will include Landsat and Sentinel imagery and improve the comparison with the recent Sentinel inventory.

How the outlines have been derived from the Lidar dDEM / hillshade and what the specific interpretation challenges are is not described (Fig. 6 in

Abermann et al. 2009 is of limited help here as it refers to a different region and boundaries are hidden by thick outlines). The authors do neither present glacier outlines as derived from the very high-resolution orthophotos nor those from a recent study using 10 m resolution Sentinel-2 data, be it for comparison or to make the case why the method they have selected is preferable. The four images in Fig. 5 are presented without such outline overlay closeups so advantages / disadvantages of the available methods under such challenging conditions are not presented. So I am unclear about what I can learn here about topic (B)?

This is a very helpful comment pointing out the necessity to improve the description in the text and visual presentation of our graphs. We can include also visual comparisons to the Sentinel data set we already cited.

[3] Assuming that Sentinel-2 data are and will be used now for quite some time to create new glacier inventories all over the world, I would have loved to see such a comparison or learn where the difficulties are at the metre scale when using Lidar hillshades compared to orthophotos. For what regions do outlines differ and by how much, how does this impact on total glacier area and what are the omission / commission errors? And how does such a best quality interpretation deviate from Sentinel-2 outlines?

We will be happy to include this in the discussion.

Such an assessment would greatly help in quantifying related uncertainties and problems when using satellite data. And, maybe even more important for this study, it would also link topic (B) to topic (A). Instead, images from 2002, 2009 and 2015 are shown with glacier outlines from 1850, 2006 and 2018 (Fig. 4). It is unclear to me what these images should reveal when their timing does not match to the outlines, when regions of difficult interpretation are not marked, and when they are so small that the important details are lost.

The time series was shown to answer the question of repeat time periods, as these time series show how quick changes take place. We see that the description of what is shown and why is weak, and we need a better link to the text.

We understand that we have to show more details, and explain better why the interpretation of imagery is so much easier based on multitemporal data, as changes happen very rapidly now.

 I suggest showing (but not in the introduction!) one image with the area changes for a larger region and then one or two close-ups illustrating

the difficulties and/or differences in interpretation.

Thank you for this clear outline of how we could significantly improve our Figures! We also suggest to rearrange this information in a separate chapter.

[4] My example (2) is about the inventory (topic A). I only find multi-temporal outlines for two sub-regions, for one of them the same outlines are repeated five times (in Figs. 4 and 9) and another one in Fig. 6B, whereas elevation changes are shown for 3 glaciers in Fig. 6(A) and for 3 + 1/5 glacier in Fig. 9(B), but using a different colour table. Where are the maps for the other glaciers? For such a small region I would expect an image covering the entire region, one with outlines and one for dhdt.

[Figure]

We will add a figure of this type with annual means of thickness change to the manuscript.

Instead, in a highly inconsistent manner Fig. 6(B) shows

additional outlines from 1969 and 2002 whereas Figs. 4 and 9 adds outlines from rock glaciers that hinder a clear interpretation of the area changes.

We can shift the rock glacier outlines to the discussion to avoid a confusion.

[5] As a suggestion, why not presenting a clear message and showing glacier extents from 1850, 1969 and 2017/18 to illustrate the dramatic shrinkage for the entire region and then use strong close-ups to illustrate the details for one or two sub-regions? I would also strongly recommend removing the rock glacier outlines from the glacier extent overlays. They are a bit distractive, refer to a different topic, and partly mark regions that seem to be too large. For example, the large one to the left in Fig. 9(A) is covering the still existing glacier 'V'. How can this be? Either there is a glacier or there is a rock glacier, but both at the same time?

We intended to point out the necessity of a homogenization of these two inventories, and at least it seems that we have been successful to show that there is such a need. We will try to improve the phrasing in a way that this point will be clearer.

I am aware that there is likely no consensus about where a debris-covered glacier ends and a possible rock glacier starts, but seeing both on top of each other is confusing. Also the LIA extent of 'H' seems to be too large as it ends beyond the 'Moränenbastion'. How can this be? If datasets were just copied from another study, I would at least discuss such inconsistencies. At this scale such interpretation differences became much more obvious.

The datasets come from other studies. We will be happy to discuss these inconsistencies! The LIA margins are based on orthophotos, and do partly deviate from the LiDAR landforms. The high uncertainty of 10 % of the LIA area in highest regions without moraines, these inconsistencies accounts not only for the uncertainties in the accumulation area, but also for the limited accuracy of mapping glacier tongue extents.

[6] This example leads me to two other weak point of the study, it seems that some of the raw data (rock glacier and LIA extents) have not been critically assessed and/or are based on different rules of interpretation. As also other terminology is used a bit loosely with reduced attention to details and the limits of interpretation*, it might be better to leave the investigation of topic (B) to another study. As an example, the study by Leigh et al. (2019) [doi.org/10.1017/jog.2019.50] provides a detailed analysis of the visibility of very small or disappearing glaciers including a classification scheme and a cross-comparison of glacier outlines using sensors of different spatial resolution and varying mapping conditions (e.g.

regarding seasonal snow). Neither this study nor the one by Janke et al. (2015) [doi.org/

10.1016/j.geomorph.2015.03.034] about the co-existence and classification of glacial and

peri-glacial landforms is mentioned here, giving me the (wrong?) impression that the authors

are unaware of the current state-of-the-art.

We focused on glaciological literature and tried to point out the development of our current definition of a glacier. But we will consider to include  the discussion on periglacial landforms.

This extends to citing glacier definitions by Walcher

(1773), Tyndall (1860) or von Klebelsberg (1948), but neither Cogley et al. (2011) nor

those from the GLIMS analysis tutorial (Raup and Khalsa 2007) which is providing guidance

for the remote sensing perspective and which I think is thus very relevant for this study.

We will include Cogley et al. (2010) and Raup and Khalsa (2010) (from the tutorial), as these are the most modern versions of the older definitions, as well as a discussion on ice cored moraines.

* For example, in L71 I read that also Schnapfenkuchl M 'cannot at first glance be identified'

(even during a field survey!) and L77 talks about increased debris cover with bare ice rarely

exposed. In fact, Schnapfenkuchl M is still a nice glacier with a well-defined accumulation

area and only very little debris cover (as visible in Figs. 2, 4 and 9A). This limited attention to

details would set for me a big question mark behind the discussion of topic (B) in this study.

It might be due to a rushed preparation of the ms, but it gives the impression that the authors

have not noticed it and are thus not in a good position to discuss the glacier question.

We are sorry for this typo! Instead of 'The three Schnapfenkuchl glaciers as they present themselves today' it should read 'The two Schnapfenkuchl glaciers V and H as they present themselves today'…

[7] As the 2nd weak point, I also miss the issue of scale-dependency of glacier interpretation,

e.g. that different glacier features are visible at different spatial resolutions, leading to a

different interpretation of glacier extent (without being wrong).

These points can be easily shown by additional images, e.g. in the discussion or the supplements.

We focused the paper on the highest spatial resolution available to detect surface elevation changes in addition to optical data.  Comparing different resolutions of optical data would be another topic, and a potential follow up including not only sentinel, but also the ASTER and LANDSAT images (how accurate can we expect the remote sensing inventories of the last decades to be, and which implications does that have on the expected uncertainties in change mapping? We are fully aware that using a different scale does not have implications on the quality of the data.

Similarly and as mentioned above when using different techniques (dDEM, hillshade, optical), the authors seem to base

their conclusions on changes that are visible only at the metre scale and maybe in the field,

but do not check this back with the relevance of such changes (or calculate the impact). To

put it sharply, why does it matter when there are a few more rocks on the glacier surface?

Can the related impact on mass balance be seen or measured at all? How can additional

glacier inventories help in keeping track of buried glacier ice (see L414/5) when the related

changes are invisible (e.g. the 'fade-out' is gradual)? What spatial resolution and repeat interval

is required to do so and what do we learn from it?

The reason to tackling buried ice is mainly the hazard potential and hydrological impacts, as stated in line 115 and 440.

In line 432 we state that

The vertical accuracy needed to represent the geomorphological processes of debris relocation depends on the steepness of the area and on the temporal interval chosen. For a period of 10 years and a slope of up to 40° for Alpine glaciers, a vertical resolution of 1/10 of the spatial resolution is sufficient to distinguish volume changes by ice melt from erosion and deposition.

We will be happy to add some examples of past disasters and event chains.

The questions asked here are excellent points for the discussion and will be added in the revised manuscript.

 What is the effort and how can this

be done in other regions?

We focus the paper on our study site, as this is a first pilot study. We suggest including surface elevation changes also in the analysis of glacier area changes of other regions where glaciers are already small and get covered by debris like presented. UAV and time lapse cams can provide solutions for small areas with potential for disasters.

I would have expected answers to such questions in this study as

the authors have excellent raw data available.

[8] The ms has also a high number of distracting inconsistencies, which give readers (at

least myself) a hard time and indicates that the ms might have been prepared in a rush. For

example, is Larrainferner in Fig. 1 marked by the number 1 (in a circle) or by the square annotated

with 'Schnapfenkuchl and Larain'?

Sorry for that, Larain was indicated wrong! The viewing direction is given by the white arrow.

We will improve the link of the Figure to the text.

[Figure]

There is no indication in the Figure caption where

this box refers to (maybe the subsets shown in Fig. 4?) or which of the glaciers are Larain or

Schnapfenkuchl H / M / V. As also the viewing direction of Fig. 2 is not shown in Fig. 1, it is

very difficult to relate the oblique aerial photo of Fig. 2 to the glaciers in Fig. 1.

[9] Similarly, presenting an extreme close-up in Fig. 3 without any reference to where this

subset is located (this comes only 16 pages later in Fig. 9) or where a location map can be

found is confusing.

Indeed we missed to give the reference of the subset to a figure later in the manuscript. We will add this in the revised version.To carry on, Fig. 4 is showing results of the study for topic A (outlines

from 2018) and Fig. 5 for topic (B), but we are still in the introduction section! Although there

could be some flexibility in the arrangement of the contents, I wonder why results are already presented in the introduction?

As mentioned above we consider to rearrange the sections for discussion the challenges of drawing the new inventory.

 I am also confused to find a diagram illustrating the

methodological workflow in Fig. 10 as a part of the Conclusions section instead of the methods

section.

We consider the workflow as aim of the study, so tried to say in line 128 of the Introduction. Therefore, it is straight forward to present the workflow in the result section. Focussing the study on the inventory results and shifting the uncertainties in the discussion, we can shift the workflow in the results section.

I would have loved to see it there to get a first overview on the methods used.

[10] This brings me to the methods section that is, in my view, a wild mix of datasets and methods. When so many different datasets (glacier outlines, DEMs, orthophotos) from different points in time, different sources and referring to different parts to the study region are used to generate the results, I think a separate 'datasets' section is fully justifiable.

We will introduce a section on datasets.

Why not arranging all data used in a compact table providing a running number, dataset type, source,

date, resolution, coverage, purpose, example figure reference, etc. This would give clear guidance and reducing confusion considerably. I think it would also help the authors in presenting the other sections in a more logical and better structured way.

Thank you for this helpful suggestion! We will arrange all the data in a Table.

[11] To finalize my general comments, I find the five tables a bit shabby. They use different styles and fonts, include empty lines, miss capitalization, place units in different rows (or not), etc., pointing again to a preparation in a rush. In this regard I recommend to arrange all tables and figures at the end of the manuscript. This avoids creating half empty pages (4, 16, 20, 24) and overflow of figure captions to the next page, and allows at the same time to have the correct figures side-by-side with the related text. Please also note that there is a possibility for tables requiring more than one page to repeat the heading line on the next page. One can also change the column width to reach an equal spacing of rows.

We will edit and rearrange the Tables to the meet required styles.

[12] Considering also my more specific comments provided below, I would conclude that this study aimed at presenting something important, but the mixture of the two topics, the unclear images and visualisation, the sloppy use of terminology, and the confusing structure makes it difficult to follow. I understand that the authors were more interested in presenting the challenges of glacier mapping and starting a discussion rather than solving the issue. However, with the data at hand I think this is a missed opportunity for providing quantitative data. With the currently missing link of the observed challenges to the derived datasets, I suggest that

the authors focus a potential revision of this study on the new inventory and related change assessment. Please show and discuss dhdt maps for the entire region (incl. CH) and reveal how outlines derived from different methods and datasets look like.

I am sorry to say that it seems that Swisstopo does not provide official LiDAR data from repeated surveys for the Swiss part of Silvretta, so that the data base for including those glaciers is missing. We are not aware of any repeat LiDAR data for the Swiss part of Silvretta from other sources. See the homepage of Swisstopo, where the repeat Lidar is restricted to areas outside the Swiss Silvretta: https://map.geo.admin.ch/

[Figure]

I see no problem in discussing

the main mapping challenges and related uncertainties in the discussion section, but

I would avoid making this a large second topic in the same study.

Thank you for this excellent suggestion to disentangle the two topics! We will consider to shift the discussion in order of the rearrangement of the sections as proposed before.

Specific comments

L7/8: How is the contents of these two sentence parts related?

Although glaciers have been retreating for decades, as you state above, this is the first time that the delineation of glaciers is not straight forward with respect to disintegration and increase of debris cover. We will rephrase the sentence to two sentences:

Glaciers are retreating since the little ice age maximum, with increasing and historically unprecedented pace during the last two decades. For the first time, the downwaste of glaciers concern the majority of glaciers even up to summit regions, necessitating the adjustment of glacier outlines at all levels from glacier tongues to steep cirque areas.

L15: '… calculated in relation to': It seems the start of the sentence is missing.

Correct, the sentence has to be removed.

L29: Although it is used widely, I would not use the term 'climate warming', largely because climate is more than temperature. 'Global warming' might work or 'warmer climatic conditions', but otherwise I would use 'atmospheric warming' here.

Correct, what we actually mean is 'climate change'.

L55: Which period, after 2000?

After 2003.

L55: Not only: try avoiding 'Not' and write it positive.

Will be changed

L60 (Fig. 1): The glacier with the number 1 should be Futschölferner rather than Larainferner

Correct, will be changed.

L63: Figs. 2 and 3 are static, they do not show any changes.

Correct, should be Figs 4 and 5.

L65-70: Why is this in the introduction? It reads like a dataset description and refers to length changes, which are not further analysed. I would likely get it when the authors would illustrate the mentioned problems with images, but instead they switch the topic and show first something rather different in Figs. 2 to 4. In Fig. 5 (L103) they come back to the topic, but the related Fig. 5 does not show where the length changes are measured to get what the problem is. I find this confusing and of little help to see the relevant points.

We will remove this paragraph.

L71: I see Schnapfenkuchl M very well in Fig. 2, why do you write that it cannot be identified at first glance?

As stated above, this should be the two Schnapfenkuchl glaciers V and H.

L72/73: What is meant with 'the geomorphological structure' and how would it look like when

it is 'dominated by ice dynamics'?

Alpine Glaciers with significant ice flow often show a change of aspect and and increasing slopeat the margins. As you suggested to include the 2 geomorphology papers, we can refer to those.

And why could this be expected for 'debris-covered valley glaciers', which are in many cases just near-stagnant down-wasting ice masses?

Because these tend to have steep lateral moraines in contrast to flat glacier tongues, so that there is a change in slope and aspect between ice and periglacial area. This is not the case here.

Though a bit hidden by the red arrow in Fig. 2, I can well see flow structures of debris

bands on the surface of Schnapfenkuchl M.

This glacier is hardly covered by debris. Shear crevasses and Ogives may be visible, but not discussed in this context. As you suggested to include geomorphology papers in the discussion, we have to adapt what you might refer to as 'sloppy wording' to correctly refer to the syntax of this discipline.

L75 (Fig. 2): The viewing angle of the image is a bit unfortunate as glaciers H and V are hard

to see from this perspective. I suggest showing a close-up - if at all - as I do not get what

the purpose of this image is (e.g. compared to Fig. 4). Moreover, 'rock glacier' is difficult

to read and the left one is also covering steep rock walls. Please do not present 'M' as a

glacier with increased debris cover and rarely exposed bare ice when the glacier is largely

free of any debris. Please also be precise with arrows, the remains of V are located

much higher up (where the snow fields are).

Ok, I had the impression that there is a glacier mouth right at the location of the arrow visible in the orthofoto of 2015, with subsidence indicated in the elevation change map. Of course this can be also classified as dead ice.

L80: Show where this subset is located in Figs. 1 or 2 or 4.

We will add this reference.

L81: Aren't these (deformed) 'stratigraphic layers' a sign of ice dynamics (see Leigh et al.

2019) and thus in contrast to the statement in L73?

This layers may be a sign of past ice dynamics. More likely they are a product of differential snow deposition and ablation resulting in the spatial distribution of firn layers.

L84 (Fig. 4): I find the white lines disturbing and would remove them, similarly for the rock

glacier outlines.

Ok, so no elevation information is needed? We can remove them, or add elevation information in a coarser scale.

We will consider to reduce the number, adapt the colour and to label the countour lines accordingly to reduce disturbance but enable orientation on elevation gradient within the figure.

The multi-temporal glacier outlines are a bit too thin.

We will adjust the line style.

Why is the glacier

name annotation not shown in the 2015 example?

No specific reasons. We will adress that.

Why do the image dates not match to

the outline dates?

As the mentioned in the method and the discussion of this study shows, it is hardly feasible to determine glacier outlines without the required high resolution surface elevation information in particular for small glacier. Thus, reasonable glacier inventories are in line with the ALS surveys and not with orthofotos, only.

As it is nearly impossible to trace the glacier changes from image to

image, I would show only one image (maybe the first or the last one) and show it big.

We will rearrange the images in order to show all necessary information.

Please also note, there is lots of glacier free area to the left of V, but Larainglacier is partly

cut away. I suggest shifting the cropping to the east.

Thank you for this comment. We will adjust the cropping accordingly.

L85: 'show that the glacier changed quite rapidly within this time': this is hard to see from the

images but also difficult to see from the outlines (as the white and orange ones interfere).

Thank you for this hint, we will change that!

L86: 'The former accumulation area of Larainferner lost contact ... and 2009': Where is this

'former accumulation area? Please mark it (and the disconnection) to make the case.

Ok, we will mark the location.

Please also note that what has been disconnected was a tributary glacier. As one can

see from the images, snow accumulation still takes place below (and to the left and right)

of this separated glacier. So the statement seems incorrect.

Ok, we will change the text to 'from the main accumulation area'.

L87: Is 'eastern Schnapfenkuchl' H? Please show close-ups, to show the temporal evolution.

Yes. We will show close ups.

L88: 2019? The image is from 2015 and the outlines are from 2018.

Sorry, we will correct the typo.

L92/93: Figure 4 does not show an image from 2006 and most of the features in the list are neither visible nor marked on the images. As mass balances have not been measured (?), the term 'accumulation areas' seems misleading and should be replaced with 'remaining snow cover' or similar. I suggest removing the englacial drainage systems from the list and consult Leigh et al. (2019) for more suitable indicators (e.g. deformed debris bands). Maybe show a close-up with H M V only to see something?

Ok, we can change that. 2006 obviously refers to Fig 5., we will check what happened here. Following the argument here, to consult Leigh, we will list all the potential glacier definition including geomorphological ones in the introduction I guess?  Close up: ok!

L95: I do not understand this, M is still a well visible larger glacier. Writing here that you would hardly map any glaciers here is highly confusing and gives the (likely wrong) impression that you have no idea how a glacier looks like, i.e. it crashes the entire study!

That is correct, we will revise this. Schnapfenkuchl V and H are nice examples for the discussion raised in this study in contrast to the more classical glacier Schnapfenkuchl M.

L96 (Fig. 5): The order of images is strange, why not clockwise or 2006 and 2012 side-byside?

We can rearrange that.

Why are there only arrows in 2006 and 2018, and where do they point to? What about marking possible glacier extents to reveal interpretation problems? As for Fig. 4, why is the glacier to the right partly cut?

We will revise this Figure and adjust the cropping.

L98: Why plural, there is only one flat summit glacier? Why 'completely disintegrates', it is still a single entity that has just lost some area?

We will rephrase this.

Where do the 'melt rates of 1.5 m' come from?

Measurements on 3 ablation stakes located on Jamtalferner. We will cite that.

Why plural? Has a melt rate been measured at several places and is the mean value?

Yes.

Where have the melt rates been measured? Is it an annual rate?

Close to Jamspitze, Chalausspitze and Ochsenjoch. Yes.

L99: I see the developing rock outcrop but where is the debris cover? Where should the debris

come from when this is a 'summit glacier'?

The debris originates from the rocks at the summit. We can include a photography of the last phase of downwaste when englacial debris spreads out as already described by Srbik for the 1930s.

L105: This seems to be incorrect. I can still see bare ice in 2012 and 2015

Even if barely ice is visible, the by far larger part is covered by debris at the glacier tongue. We will improve the description.

 (what is with

2018, is the ice back then?).

We cannot follow this idea, but may add a sentence on that.

 The main problem here is that Fig. 5 does not reveal where

the length change measurements are taken or which parts of the glacier turn from bare

ice to debris-covered ice.

Ok, we can include this data.

Apart from this, I disagree with the interpretation by the authors:

There was likely some ice left in ice cored moraines in 2012, but by 2018 all ice melted.

We still have subsidence and can observe loose ground in the area. We no indication that the ice melted completely in 2018.

So this is not really a glacier showing an increasingly debris-covered tongue, but glacier

ice that melted, leaving behind bare (debris covered) ground.

The debris on this glacier tongue mainly originates from englacial debris accumulating at the glacier surface while ice downwasting. This reliable finding is based on field trips to this area. However, if you have evidence on your thesis from field visit or literature we are open to that discussion..

L106: As this glacier has barely any debris cover, the impact on the accuracy is likely small.

In any case, a comparison to satellite-derived extents from 2016 would have been worthwhile

to make the point.

We will include Sentinel data in the revised version and use that as a best example.

L107: In the caption I read 1.5 m? #

Yes, more than one is no contradiction to 1.5 and accounts for the range within the measurements.

Does the 'as measured on Jamtalferner' belong to the

'more than 1 m w.e.'? Why is it here?

As above, same stakes.

Now it reads as if the disintegration of the flat glacier

has been measured on Jamtalferner, which I find confusing.

We will rephrase that.

L107: As stated above, this is likely a down-wasting but not (yet) a disintegrating glacier that

should be composed of several individual pieces. Moreover, disintegration has only a limited

impacted on mapping accuracy, so this example does not fit very well.

We will rephrase that.

L108: Should this mean 'debris from rock fall'? Where should the debris come from during

disintegration or down-wasting? Is this all englacial debris?

We so far did not find a way to tackle the source, it seems that also subglacial sediments are
transported to the surface by melt water if the ice thickness is lower that about 5 m.

L110: Why 'we will ...in the coming decades'? This sounds as if this might happen at some

point in the future. Actually, we follow it for more than 3 decades now (Figs, 4, 5 6b, 9a).

… even 150 years, but most of the glaciers in the study did enter a new phase now.

L110: What is a 'large glacier system'?

Sorry, we will add an example what can be considered large in this respect, as for example
Bieltalferner 1850.

L111: 'for the last hundred years': Again, a rather imprecise statement considering the well

documented advance phases of the glaciers over this time period. The related moraines

from the 1920s and 1980s advance can also be seen on some of the orthophotos.

We will include a citation on these advances.

Similarly,

this is not only observed in the Silvretta region but in the entire Alps (and globally).

We can add citations on other studies here.

L110-112: I would like to use the above example for pointing to the often confusing / interrupted

/ hard to grasp structure of the text. Why first writing that rapid recession will likely

occur in the future and then stating that it has already happened over the past 100 years?

The second sentence refers to the second part of the first sentence. We will go over the text and shorten and didentangle the sentences to improve the flow of the text.

The first statement implies that it has not happened yet so the next sentence is a direct confirmation that the first statement is wrong. Why not write that widespread glacier recession was observed in the Alps over the past 150 years (incl. Silvretta), that recession has more recently accelerated (this would require some proof of evidence) and will very likely continue in the future (e.g. due to the committed mass loss)?

Thank you for this suggestion!

L 113: The 5 pages before only discuss aerial photography, the advantages of Lidar are not discussed at all. The possibility to detect hidden ice by its down-wasting is not even mentioned.

Ok, thank you for this hint!

There is also nothing about the spatial resolution requirements to detect the relevant details.

When we include the close ups, we will give special attention to that.

 In other words, the introduction discusses a lot of things but has a limited perspective on what is relevant or motivating the study. Please introduce all relevant topics appropriately, this is in my opinion the purpose of an introduction section.

ok

L113: Where is the 'glacier fade out' by Lidar been shown? I only see two distinct maps for different regions and a fixed time interval. 'Fade out' is for me a process occurring over a longer time and should thus be documented by time series. This has been done for outlines, but not for Lidar as stated here. Please check the text against the work presented.

Ok, we will include that volume changes include changes between two different dates. We do not use single LiDAR images for mapping glacier outlines.

L114/5: This statement is correct but what has it to do with the text before, e.g. the difficult mapping? What is the relevance of the glaciers presented in Figs. 2 - 5 for sea-level rise?

That is correct, we will remove the citation.

L115: Instead of using glacial and post-glacial (which more refer to LGM glacier extents) I suggest writing 'a landscape with glaciers evolving into a glacier-free landscape'.

ok

L116: How does the monitoring by remote sensing help in 'dating of paleoglacial landforms'?

For example by constraining the time period of succession in the periglacial areas, which is needed for the interpretation of pollen profiles.

L118: I think this is not the definition of a glacier inventory.

We will change that to 'Austrian glacier inventory' and replace 'definition' by 'aim'.

L118: 'of all glaciers in the region': I do not find a figure showing this

Figure 1?

and also Table 5 is only

presenting glacier areas from two points in time rather than discussing the changes.

We can include LIA, 1969 and 1998.

L119/20: These are indeed two relevant questions, but where has this been shown in the

study? Which landforms can be neglected and what is the impact?

We will quantify that.

L120: 'in a stage of rapid glacier shrinkage'

ok

L 126 (bullet II): I miss a critical discussion if the short 2002 to 2004/06 interval makes sense

at all, e.g. compared to uncertainties.

This means a bit changing the focus of the paper from the first repeat LiDar inventory to more basic questions treated in Fischer et al 2015. This data exists, and there is no need to analyse the data of this short period. Nevertheless, as the extreme summer of 2003 is included, it is still interesting to discuss this short period.

There is also no image showing a dhdt map over

this time period.

Without such a proper documentation and discussion, the presented results

for this period seem unreliable and should thus not be presented. If the authors think

the comparison is reliable and reveals something important, please show it and discuss it!

As we have a number of additional task requested in this review, I think we better skip the time series of inventories and include the remote sensing comparison as above.

L128: 'under conditions of rapid glacier decay'

ok

L131: As suggested in the general comments, please use a separate section to describe the

datasets and the methods. Please consider discussing in the introduction only the larger

background and motivation for the study, before presenting details of the study region in a

further section. And please keep out the results (Figs. 4 / 5) from the introduction!

Ok, as answered above

L133-136: All these datasets are shown in Section 1 before they are introduced …

We will restructure the text.

L169 (Fig. 6A): The bluish parts in the background (with positive values) look like a shaded

relief, indicating that the two DEMs used here have not been correctly co-registered.

We described the coregistration process with all uncertainties in detail, including the difference in
resolution of the images. The deviations in the image just show these uncertainties well known and
described for steep terrain.

What does N, M, S mean?

Nord/North (N) Mitte/Middle (M) Süd/South (S)

It is unclear to me why the outlines have been mapped where

they are, what are the rules? I can follow the 'maximum change' rule near the terminus,

but partly the line is outside the bluer regions.

Ok we will check that.

L169 (Fig. 6B) I wonder why this figure requires outlines from 4 further dates, hiding details

of what is visible in the image? When the purpose is to reveal the accuracy of the dhdt interpretation

in Fig. 6A, I would show close-ups, compare it against outlines derived from

the orthophoto, and discuss differences in interpretation at the pixel level. The current

overlay tells me nothing in this regard.

ok

L180-250: This is all rather theoretical and should be there, but I am unclear how it helps

interpreting the results or how this information is supporting the goals of the study. I have

described above and in the general comments what I would have liked to see here (e.g. a

comparison of the results obtained with different methods or how the differences in interpretation

impact on the overall results).

This part is important, as it explains why you see the bluish color above.

L220: Shouldn't this be Section 2.5?

yes

L250-290: The results section is only presenting inventory data and volume changes. The

results for the two research questions (L119/20) are missing. There is a qualitative discussion

in Section 4, but no quantitative evidence is given to support the statements.

Ok, we have to adapt the study aims to include all the topics you suggested.

L285: Why should there be a relation between mass balance and glacier area?

We have been interested to know how the specific glacier of different sizes reacted to climate change. We do not propose a relation, it is just that the last inventories (in Austria as well as Switzerland) showed different responses of small and large glaciers, mainly because the tongues of the large glaciers are subject to large ablation rates.

 Wouldn't be

glacier mean slope, aspect or elevation the more interesting variables for a scatter plot?

We will discuss to include that.

L288: Please sort the table alphabetically or refer to the numbers in Fig. 1. Mark ALL glaciers

listed in this table in Fig. 1 (rather than numbering a selection that is never used for

anything). This Table might be shown in the supplement.

ok

L306: I think ponds or crevasses are not 'stable surface structures'.

We will add the time we refer to.

L308: I do not understand this connection, what does a bore hole tell me about surface velocities

due to sliding debris and rock?

The vertical profile of velocity and the stratigraphy allows to see which stratigraphic layers of the rock glaciers move relative to others

And why 'sliding' on the ice?

We refer here to the deformation profile of Lazaun rock glacier , Krainer et al- see the draft.

Isn't a rock glacier a

frozen body of ice and rock (below zero degrees) that is creeping?

Yes. However, rock and ice are not evenly distributed within the matrix , and it can contain ice lenses.

L309: Should this mean exits or emerges at the terminus? Does this emerging water define

where the terminus is?

At least we take the samples where we observe the current outlet of the ice body.

L315: This might be correct but what about the effort and the limits imposed by spatial resolution?

When resources are limited and aerial photography or Lidar DEMs are unavailable,

one has to decide what to do with the available personnel resources and datasets to

get an inventory finalized. I would have expected here a recommendation about such limits

from the datasets analysed here.

There is a clear if, 'if we want to keep track'. Limits are found a bit further in the text, and as you suggested we can include close ups so that it is easier for the reader to see the limitations.

L316: This should be Section 4.2 rather than 4.4 and so forth for 4.5 (=> 4.3), etc.

Yes, as above.

L322: How can ice melt in a region with permafrost, i.e. with temperatures below zero?

Why not? We often observe permafrost in a suited microclimatic setting neighboring melting snow and ice or even running water.

L323: Why debris flows? This looks as ice that is exposed because rocks slid down on a

steep slope (cf. Fig. 3).

This often happens during heavy precipitation events.. Nevertheless, debris flow (not in a hydrological sense) may be the wrong wording here, changed to debris sliding down the ice.

L325: 'needs a high temporal frequency of inventory data' I think this issue cannot be resolved

by just increasing the frequency of inventory data. The temporal and spatial scale

we are discussing here points more to a frequent (daily?) observation by a good webcam

with repeat DEMs derived from drones.

Sorry we have no data to compare to the annual or even subseasonal time scale. We argue that at least every 5 to 10 years a high resolution inventory will be needed to keep track.

 Such an effort can likely be justified for scientific

investigations, but for regular monitoring it seems to be too demanding.

This may depend on the hazard potential and the corresponding area of interest.

L334: The inserts show some bare ice but where are the debris flows?

We will add a close up.

L340ff: What about the remote sensing based glacier definition from GLIMS?

See above, will be included.

L361: Just that two groups of scientists interpreted the landscape differently does not mean

that a transformation of the mapped forms has taken place.

Correct, and try to state here that it is a different interpretation of the same landform.

In a region with permafrost

and steep slopes the surface can also start creeping without the ice from a former glacier.

correct

Wouldn't also the glaciers temperature regime require a change from sliding to creeping?

Up to now there are  no thermistor chains available at the tongues, nor measurements of sliding velocities.

L386. What about comparing mass balances to Silvretta Glacier?

We tried to focus on the Austrian inventory.

L424: Why is Fig. 10 not in the methods section?

Because we consider this Figure to be a result.We can also shift the Figure in the extra section on mapping.

---

## Author Comment (AC3)

Response to RC #3 from 14 May 2021 https://tc.copernicus.org/preprints/tc-2020-376/

We thank reviewer #3 for their comments and suggestions which will help us to improve the manuscript!

Review in black letters

Response in blue letters

This manuscript is an interesting and well written paper about the evolution of small-scale glaciers in a sub-region of the European Alps. Disintegration and especially steady covering by rock debris creates problems in mapping glaciers and creating glacier inventories.

The authors describe very well the material and methods and have a thorough discussion at the end. Interesting is the assumption of an updated definition of the term glacier.

What about glacier movement in the definition? Can that be discarded?

> We thank the reviewer for their interest! The discussion on existing ice dynamics is important, as stagnant ice bodies of larger sizes can only exist as transient states (and will melt soon under current conditions if not covered by thick debris layers). We showed that the increase or decrease of ice flow velocity is one of the earliest responding indicator of glacier state (Stocker Waldhuber et al., 2019 https://essd.copernicus.org/articles/11/705/2019/). Including dynamics in a definition would mean to exclude stagnant ice bodies, and include them again if they start to move again (which can happen within a season). As in the moment we cannot measure dynamics of a buried ice body without drilling, all arguments point towards discarding the ice movement in the definition of a glacier, with the result of including various types of stagnant and buried ice.

> We see that this point needs some work and will elaborate the point on ice motion.

Minor comments:

L254: … the latest period was 2.4%, which is … (you wrote loss; therefore, it should be a positive number)

> We will change that throughout the paper.

Table 3: should be -17km² area change in 1969

> Thank you for pointing out this typo, we will change that.

L266ff (and further in the text): please check terms highest/lowest/maximum etc. à they should all be the other way round as you refer to negative numbers

> As for L254, we will change that.

L270f: …, reducing the overall volume loss as no ice to melt is left in the areas with highest ablations in the past.--> this sentence is not clear to me.

> This sentence is definitely too long and will be rephrased. At Jamtalferner, the areas at the glacier tongues where more that 6 m ice ablation was measured are ice free now. Now only the areas with about 4 m of ablation in the past are left to contribute to volume change. Therefore, despite an increase in ablation at individual stakes, total volume loss decreased.

Table 5: I would suggest a map with different colors/symbols instead of the table.

> As one of the previous reviewers suggested to move Table 5 to the supplements, we will be happy to add a map as suggested in the main text.

L319ff: Could another reason be a change in debris cover?

> Of course, thank you, we will mention that.

L375: does away with à better use overcomes?

> We will rephrase that.

L391: Caucasus comparable to Silvretta?

> We will include a few comments on the difference of these mountain regions, sorry not to have mentioned that, as it seemed very clear to us.

L419: … the future

> We will change that.

Fig. 10: I do not understand how you discriminate between volume change? and debris accumulation dominant? following the thickness change. There are two arrows without further indication.

> Thank you for pointing out this two issues. There is only one experiment regarding debris accumulation on ice (with the help of net) on a single location where debris comes from the rock faces, so that in the moment we see no way to estimate the amount of debris accumulation (we will work on that!). The other problem is that there is now way to distinguish 'external' debris from englacial one coming to the surface. Here we are not sure how to design an observation procedure.

> We will rethink the arrows from the thickness change, this definitely needs explanation.

L460: The glacier inventory data is stored in https://doi.org/10.1594/PANGAEA.844988.

**Citation**: https://doi.org/10.5194/tc-2020-376-RC3

> We will change that and include the citation

---

## Author Response (AR1)

Response to **RC1**: 'Comment on tc-2020-376', Anonymous Referee #1, 16.07. 2021

RC1

Response 16.07.2021

This is well written and interesting paper that describes glacier disintegration in a region in Austria based on an extensive data set of DEMs and orthophotos. The glaciers in question are all rather small (a few km² or smaller) and some have disintegrated into partly or fully debris-covered patches. The authers point out the need for a scientific discussion about the proper aim and suitable methodology for monitoring small, totally debris-covered cryogenic geomorphological structures that are remnants of recently active glaciers. At some spatial scale and degree of debris-cover, monitoring of such remnants might not be within the proper scope of glacier monitoring as such but should be considered as a task for monitoring and studies of rock glaciers and permafrost. Setting a hard threshold in terms of the size of very small glaciers for including them in glacier inventories and considering then glacier monitoring is, however, not straightforward.

We thank reviewer 1 for their very constructive and helpful comments.

Comments:

p. 14, l. 246: The authors use the area at the first point in time $A(t_1)$ to compute the area-averaged specific mass balance $b_{geo}$. I would have thought that the average of the areas at the beginning and end of the period $(A(t_1)+A(t_2)/2$ would be more suitable? This would be consistent with the description on p. 92 in the IACS glossary of glacier mass balance (Cogley et al., 2011).

We added $b_{geo}$ calculated with averaged areas and shifted both versions of the table plus the resulting differences to the supplements. Comparing the two different ways calculating the geodetic balance revealed interesting results.

p. 22, l. 340-353: It might be worthwhile to add the IACS (Cogley et al., 2011, p. 45) and GLIMS (Raup and Khalsa, 2010, p. 4) definitions of glaciers in this list of definitions of the term "glacier" since these have and "official" status in the glaciological community.

We included:

*"A perennial mass of ice, and possibly firn and snow, originating on the land surface by the recrystallization of snow or other forms of solid precipitation and showing evidence of past or present flow… In contrast to what is natural in dynamic glaciology and glacial geomorphology, for mass-balance purposes the glacier consists only of frozen water. Sediment carried by the glacier is deemed to be outside the glacier."*

Cogley et al., 2011

and the GLIMS definition of glaciers

*"A glacier or perennial snow mass, identified by a single GLIMSglacier ID, consists of a body of ice and snow that is observed at the end of the melt season, or, in the case of tropical glaciers, after transient snow melts. This includes, at a minimum, all tributaries and connected feeders that contribute ice to the main glacier, plus all debris-covered parts of it. Excluded is all exposed ground, including nunataks. …. A stagnant ice mass still in contact with a glacier is part of the glacier, even if it supports an old-growth forest…All debris-covered parts of the glacier must be included…. Rock glaciers and heavily debris-covered glaciers tend to look similar, but their geneses are different. GLIMS does not currently deal with the former, but does include the latter."*

Raup, B. and Khalsa, S.J.S., 2010. GLIMS Analysis Tutorial. GLIMS, Global Land IceMeasurements from Space, NSIDC, www.GLIMS.org
https://www.glims.org/MapsAndDocs/assets/GLIMS_Analysis_Tutorial_a4.pdf

Regarding the definition of "glacier" on p. 22 and 23, it might be mentioned that definition (1), that includes dead and buried glacier ice, would also include ice-cored morains, even with deeply buried ice that may last centuries, which would presumably greatly complicate the creation and management of glacier inventories.

Yes, this is correct. We added this point to the discussion and added one example of dead ice.

Minor and editorial comments:

The superscipted power of 2 in "km²" is missing in the pdf I am reading (many places).

We have to check this in the next proof-reading version.after upload-> in our version, downloaded from the TCD server, km² is shown correctly, in Tables and text, eg.

From the LIA maximum to 2017/18, the Austrian Silvretta lost 68% of its glacier area (Table 3). The mean annual area loss in the latest period was -2.4%, which is more than twice the loss of the period before. The 10 totally debris-covered glaciers cover an area 0.303 km² (Table S3). For three glaciers (ID 13006, Fluchthornferner S and Litzner Gletscher E), neither bare ice nor signs of motion or a drainage system were visible, mean thickness changes between 2004/06 and 2017/18 were smaller than 2.6 m without a clear thickness change pattern to indicate an ice margin. From that we can conclude that the subsurface ice merely melted.

Area changes were calculated for all glaciers for the periods LIA maximum -1969 and 1969-2002 (Table 3). The dates of LiDAR campaigns differ between the federal states of Vorarlberg (2004 and 2017) and Tyrol (2006 and 2018). Later periods are thus 2002-2004/06, and 2004/06-2017/18.

**Table 3: Glacier area in the Austrian part of the Silvretta between LIA maximum, 1969, 2002, 2004/06 and 2017/18. For Vorarlberg, LiDAR flights were carried out in 2004 and 2017, for Tyrol in 2006 and 2018.**

| Time | Area | % of LIA area | Period | Area change | | |
|---|---|---|---|---|---|---|
| | km² | | | km² | % | % per yr |

As a second reviewer reported the problem, we might address that in the letter to the editor, as it is not possible for us to solve problems related to specific libraries.

The "-" hyphen in negative numbers and number ranges should be changed to en-dash or a minus sign ("--" in LaTeX), for example "2002-2004" should be "2002--2004"

The manuscript is written in word, negative numbers have been already written with a minus sign, for periods we replaced the minus sign by {CTRL -} .

p. 12, l. 197: "between 1000 to-2000 m" --> "between 1000 to 2000 m"

corrected

p. 15, l. 269: "lower than" --> "lower in magnitude than"

corrected

p. 19, l. 305: "... if there is any ice left." --> "... whether there is any ice left."

corrected

p. 20, l. 315: "Without keeping them in inventories, we lose track of these transient states." --> "We lose track of these transient states by dropping these glaciers from the inventories."
changed

p. 24: "The annual change rates of area and volume change indicate ..." --> "The annual rates of change of area and volume indicate ..."
corrected

p. 26: "The glacier inventory data in https://doi.org/10.1594/PANGAEA.844988" --> "The glacier inventory data are available at https://doi.org/10.1594/PANGAEA.844988"

corrected

We thank reviewer RC 2 for their very constructive and detailed review which will clearly help to improve this paper by pointing out weaknesses, but also suggesting the respective improvements which we will be necessary to be included for final publication. Please find all the details below.

Review of TC-2020-376 by Fischer et al.

[1] The study by Fischer et al. is presenting for glaciers in the Austrian part of the Silvretta

group two different topics: (A) the results of a new glacier inventory and glacier changes (in

area and mass) over time and (B) the problems in accurately collecting the required information

in times of massive glacier down-wasting and disintegration. Both topics are relevant

and of interest, but I miss the connection between them and a more detailed elaboration of

cause and effect that goes beyond the presentation of new numbers in tables and some

vague speculations about processes that are not presented or observed.

We thank for pointing out that the interconnection of these two topics is not clear, and that we have to address this more directly than it is done in the first version of the manuscript:

The interconnection results from the fact that we have to address the problems related to the glacier mapping under the given glacier conditions, and that the result are increasingly uncertain.

We rephrased and restructured the article to make the observation of processes clearly visible by presenting less glaciers, but more of the information in the time series we used.

[2] To give two examples here: (1) The difficult identification of glaciers under debris cover is

a key aspect of the study (topic B), but only the results of one method (elevation changes

from a Lidar dDEM) are shown.

We intended to show that surface elevation change data is essential in addition to optical information for a proper delineation in particular of debris covered glacier parts. Thus, it is not a single method applied, rather an additional method to standard process of delineating glaciers from lidar hillshades, orthophotos or satellite images only. I presume that the idea that we compare mapping results from orthophotos with LiDAR stems from the abstract, where we found and removed a misleading information. As the orthophotos are available for other years than the LiDAR, there were used for a plausibility check only.

But following this argument here, we will improve the comparison to other methods of mapping. We will include Landsat and Sentinel imagery and improve the comparison with the recent Sentinel inventory.

How the outlines have been derived from the Lidar dDEM /

hillshade and what the specific interpretation challenges are is not described (Fig. 6 in

Abermann et al. 2009 is of limited help here as it refers to a different region and boundaries

are hidden by thick outlines).

The authors do neither present glacier outlines as derived from

the very high-resolution orthophotos nor those from a recent study using 10 m resolution

Sentinel-2 data, be it for comparison or to make the case why the method they have selected

is preferable.

We presumed that the general methods are well known, as published more than 10 years ago. But as it seems helpful, we will include some more basic Figures. We added a comparison to Sentinel- 2 outlines, and added Landsat 8 images.

The four images in Fig. 5 are presented without such outline overlay closeups

so advantages / disadvantages of the available methods under such challenging conditions

are not presented. So I am unclear about what I can learn here about topic (B)?

This is a very helpful comment pointing out the necessity to improve the description in the text and visual presentation of our graphs. We included visual comparisons to the Sentinel data set we already cited. Generally, the aim of the article was not to present mapping techniques, but rather present the new challenges resulting from the glacier state.

[3] Assuming that Sentinel-2 data are and will be used now for quite some time to create

new glacier inventories all over the world, I would have loved to see such a comparison or

learn where the difficulties are at the metre scale when using Lidar hillshades compared to

orthophotos. For what regions do outlines differ and by how much, how does this impact on

total glacier area and what are the omission / commission errors? And how does such a best

quality interpretation deviate from Sentinel-2 outlines?

We are happy to include Sentinel-2 data in the presentation and in the discussion.

Such an assessment would greatly

help in quantifying related uncertainties and problems when using satellite data. And, maybe

even more important for this study, it would also link topic (B) to topic (A). Instead, images

from 2002, 2009 and 2015 are shown with glacier outlines from 1850, 2006 and 2018 (Fig.

4). It is unclear to me what these images should reveal when their timing does not match to

the outlines, when regions of difficult interpretation are not marked, and when they are so

small that the important details are lost.

The time series was shown to answer the question of repeat time periods, as these time series show how quick changes take place. We see that the description of what is shown and why is weak, and we need a better link to the text.

We understand that we have to show more details, and explain better why the interpretation of imagery is so much easier based on multitemporal data, as changes happen very rapidly now.

 I suggest showing (but not in the introduction!) one

image with the area changes for a larger region and then one or two close-ups illustrating

the difficulties and/or differences in interpretation.

Thank you for this clear outline of how we could significantly improve our Figures! We also suggest to rearrange this information in a separate chapter.

[4] My example (2) is about the inventory (topic A). I only find multi-temporal outlines for two

sub-regions, for one of them the same outlines are repeated five times (in Figs. 4 and 9) and

another one in Fig. 6B, whereas elevation changes are shown for 3 glaciers in Fig. 6(A) and

for 3 + 1/5 glacier in Fig. 9(B), but using a different colour table. Where are the maps for the

other glaciers? For such a small region I would expect an image covering the entire region,

one with outlines and one for dhdt.

[Figure]

We will add a figure of this type with annual means of thickness change to the manuscript.

Instead, in a highly inconsistent manner Fig. 6(B) shows

additional outlines from 1969 and 2002 whereas Figs. 4 and 9 adds outlines from rock glaciers

that hinder a clear interpretation of the area changes.

We can shift the rock glacier outlines to the discussion to avoid a confusion.

[5] As a suggestion, why not presenting a clear message and showing glacier extents from

1850, 1969 and 2017/18 to illustrate the dramatic shrinkage for the entire region and then

use strong close-ups to illustrate the details for one or two sub-regions? I would also strongly

recommend removing the rock glacier outlines from the glacier extent overlays. They are a

bit distractive, refer to a different topic, and partly mark regions that seem to be too large.

For example, the large one to the left in Fig. 9(A) is covering the still existing glacier 'V'. How

can this be? Either there is a glacier or there is a rock glacier, but both at the same time?

We intended to point out the necessity of a homogenization of these two inventories, and at least it
seems that we have been successful to show that there is such a need. We will rephrased the text.

I am aware that there is likely no consensus about where a debris-covered glacier ends and a

possible rock glacier starts, but seeing both on top of each other is confusing. Also the LIA

extent of 'H' seems to be too large as it ends beyond the 'Moränenbastion'. How can this

be? If datasets were just copied from another study, I would at least discuss such inconsistencies.

At this scale such interpretation differences became much more obvious.

The datasets come from other studies. The LIA margins are based on orthophotos, and do partly
deviate from the LiDAR landforms. The LIA data set published comes with an an uncertainty of 10 %
of the area, and is not meant to fit modern LiDAR data (see Fischer et al., 2015).

[6] This example leads me to two other weak point of the study, it seems that some of the

raw data (rock glacier and LIA extents) have not been critically assessed and/or are based

on different rules of interpretation. As also other terminology is used a bit loosely with reduced

attention to details and the limits of interpretation*, it might be better to leave the investigation

of topic (B) to another study. As an example, the study by Leigh et al. (2019)

[doi.org/10.1017/jog.2019.50] provides a detailed analysis of the visibility of very small or

disappearing glaciers including a classification scheme and a cross-comparison of glacier

outlines using sensors of different spatial resolution and varying mapping conditions (e.g.

regarding seasonal snow). Neither this study nor the one by Janke et al. (2015) [doi.org/

10.1016/j.geomorph.2015.03.034] about the co-existence and classification of glacial and

peri-glacial landforms is mentioned here, giving me the (wrong?) impression that the authors

are unaware of the current state-of-the-art.

Thank you pointing out the two papers, which address very different topics. We are not mainly focusing on landform classification, but just pose the question when glaciers should be skipped from an inventory. We rephrased and restructured the manuscript to make that clearer. Nevertheless, we cited both articles to make differences in the approaches clearer.

This extends to citing glacier definitions by Walcher

(1773), Tyndall (1860) or von Klebelsberg (1948), but neither Cogley et al. (2011) nor

those from the GLIMS analysis tutorial (Raup and Khalsa 2007) which is providing guidance

for the remote sensing perspective and which I think is thus very relevant for this study.

We included Cogley et al. (2010) and Raup and Khalsa (2010) (from the tutorial), as these are the most modern versions of the older definitions, as well as a discussion on ice cored moraines.

* For example, in L71 I read that also Schnapfenkuchl M 'cannot at first glance be identified'

(even during a field survey!) and L77 talks about increased debris cover with bare ice rarely

exposed. In fact, Schnapfenkuchl M is still a nice glacier with a well-defined accumulation

area and only very little debris cover (as visible in Figs. 2, 4 and 9A). This limited attention to

details would set for me a big question mark behind the discussion of topic (B) in this study.

It might be due to a rushed preparation of the ms, but it gives the impression that the authors

have not noticed it and are thus not in a good position to discuss the glacier question.

We are sorry for this typo! Instead of 'The three Schnapfenkuchl glaciers as they present themselves today' it should read 'The two Schnapfenkuchl glaciers V and H as they present themselves today'…

[7] As the 2nd weak point, I also miss the issue of scale-dependency of glacier interpretation,

e.g. that different glacier features are visible at different spatial resolutions, leading to a

different interpretation of glacier extent (without being wrong).

These points can be easily shown by additional images, e.g. in the discussion or the supplements.

We focused the paper on the highest spatial resolution available to detect surface elevation changes in addition to optical data.  Comparing different resolutions of optical data would be another topic, and a potential follow up including not only Sentinel-2, but also the ASTER and Landsat satellites as

data base to assess the accuracy of the remote sensing inventories of the last decades, and to analyse  the impact on the expected uncertainties in glacier change mapping.

Similarly and as mentioned above when using different techniques (dDEM, hillshade, optical), the authors seem to base

their conclusions on changes that are visible only at the metre scale and maybe in the field,

but do not check this back with the relevance of such changes (or calculate the impact). To

put it sharply, why does it matter when there are a few more rocks on the glacier surface?

Can the related impact on mass balance be seen or measured at all? How can additional

glacier inventories help in keeping track of buried glacier ice (see L414/5) when the related

changes are invisible (e.g. the 'fade-out' is gradual)? What spatial resolution and repeat interval

is required to do so and what do we learn from it?

The reason to tackling buried ice is mainly the hazard potential and hydrological impacts, as stated in line 115 and 440.

The vertical accuracy needed to represent the geomorphological processes of debris relocation depends on the steepness of the area and on the temporal interval chosen. For a period of 10 years and a slope of up to 40° for Alpine glaciers, a vertical resolution of 1/10 of the spatial resolution is sufficient to distinguish volume changes by ice melt from erosion and deposition.

What is the effort and how can this be done in other regions?

We focus the paper on our study site, as this is a first pilot study.

I would have expected answers to such questions in this study as

the authors have excellent raw data available.

[8] The ms has also a high number of distracting inconsistencies, which give readers (at

least myself) a hard time and indicates that the ms might have been prepared in a rush. For

example, is Larrainferner in Fig. 1 marked by the number 1 (in a circle) or by the square annotated

with 'Schnapfenkuchl and Larain'?

Sorry for that, Larain glacier was indicated wrong! The viewing direction is given by the white arrow.

We will improve the link of the Figure to the text.

[Figure]

There is no indication in the Figure caption where

this box refers to (maybe the subsets shown in Fig. 4?) or which of the glaciers are Larain or

Schnapfenkuchl H / M / V. As also the viewing direction of Fig. 2 is not shown in Fig. 1, it is

very difficult to relate the oblique aerial photo of Fig. 2 to the glaciers in Fig. 1.

[9] Similarly, presenting an extreme close-up in Fig. 3 without any reference to where this

subset is located (this comes only 16 pages later in Fig. 9) or where a location map can be

found is confusing.

We corrected the Figure.

To carry on, Fig. 4 is showing results of the study for topic A (outlines

from 2018) and Fig. 5 for topic (B), but we are still in the introduction section! Although there

could be some flexibility in the arrangement of the contents, I wonder why results are already
presented in the introduction?

We restructured the manuscript and removed the glacier margins of 2018 from the first Figures.

I am also confused to find a diagram illustrating the

methodological workflow in Fig. 10 as a part of the Conclusions section instead of the methods

section.

We consider the workflow as aim of the study, so tried to say in line 128 of the Introduction.
Therefore, it is straight forward to present the workflow in the result section.

I would have loved to see it there to get a first overview on the methods used.

[10] This brings me to the methods section that is, in my view, a wild mix of datasets and methods. When so many different datasets (glacier outlines, DEMs, orthophotos) from different points in time, different sources and referring to different parts to the study region are used to generate the results, I think a separate 'datasets' section is fully justifiable.

We splitted the section into data and methods.

Why not arranging all data used in a compact table providing a running number, dataset type, source,

date, resolution, coverage, purpose, example figure reference, etc. This would give clear guidance and reducing confusion considerably. I think it would also help the authors in presenting the other sections in a more logical and better structured way.

Thank you for this helpful suggestion! We arranged all the data in a Table.

[11] To finalize my general comments, I find the five tables a bit shabby. They use different styles and fonts, include empty lines, miss capitalization, place units in different rows (or not), etc., pointing again to a preparation in a rush. In this regard I recommend to arrange all tables and figures at the end of the manuscript. This avoids creating half empty pages (4, 16, 20, 24) and overflow of figure captions to the next page, and allows at the same time to have the correct figures side-by-side with the related text. Please also note that there is a possibility for tables requiring more than one page to repeat the heading line on the next page. One can also change the column width to reach an equal spacing of rows.

We edited the Tables to the meet required styles, there seems to be a problem with the pdf, creating missing subscripts in e.g. km².

[12] Considering also my more specific comments provided below, I would conclude that this study aimed at presenting something important, but the mixture of the two topics, the unclear images and visualisation, the sloppy use of terminology, and the confusing structure makes it difficult to follow. I understand that the authors were more interested in presenting the challenges of glacier mapping and starting a discussion rather than solving the issue. However, with the data at hand I think this is a missed opportunity for providing quantitative data. With the currently missing link of the observed challenges to the derived datasets, I suggest that the authors focus a potential revision of this study on the new inventory and related change

assessment. Please show and discuss dhdt maps for the entire region (incl. CH) and reveal

how outlines derived from different methods and datasets look like.

I am sorry to say that it seems that Swisstopo does not provide official LiDAR data from repeated surveys for the Swiss part of Silvretta, so that the data base for including those glaciers is missing. We are not aware of any repeat LiDAR data for the Swiss part of Silvretta from other sources. See the homepage of Swisstopo, where the repeat Lidar is restricted to areas outside the Swiss Silvretta: https://map.geo.admin.ch/

[Figure]

I see no problem in discussing

the main mapping challenges and related uncertainties in the discussion section, but

I would avoid making this a large second topic in the same study.

Thank you for this excellent suggestion to disentangle the two topics! We will consider to shift the discussion in order of the rearrangement of the sections as proposed before.

Specific comments

L7/8: How is the contents of these two sentence parts related?

Although glaciers have been retreating for decades, as you state above, this is the first time that the delineation of glaciers is not straight forward with respect to disintegration and increase of debris cover. We will rephrase the sentence to two sentences:

Glaciers are retreating since the little ice age maximum, with increasing and historically unprecedented pace during the last two decades. For the first time, the downwaste of glaciers concern the majority of glaciers even up to summit regions, necessitating the adjustment of glacier outlines at all levels from glacier tongues to steep cirque areas.

The abstract was restructured, to present the mapping uncertainty as part of the discussion.

L15: '... calculated in relation to': It seems the start of the sentence is missing.

Correct, the sentence was removed.

L29: Although it is used widely, I would not use the term 'climate warming', largely because

climate is more than temperature. 'Global warming' might work or 'warmer climatic conditions',

but otherwise I would use 'atmospheric warming' here.

Correct, what we actually mean is 'climate change', referring to changes in patterns of atmospheric circulation.

L55: Which period, after 2000?

After 2003.

L55: Not only: try avoiding 'Not' and write it positive.

Changed and rephrased.

L60 (Fig. 1): The glacier with the number 1 should be Futschölferner rather than Larainferner

Correct, changed.

L63: Figs. 2 and 3 are static, they do not show any changes.

Correct, rephrased.

L65-70: Why is this in the introduction? It reads like a dataset description and refers to

length changes, which are not further analysed. I would likely get it when the authors

would illustrate the mentioned problems with images, but instead they switch the topic

and show first something rather different in Figs. 2 to 4. In Fig. 5 (L103) they come back

to the topic, but the related Fig. 5 does not show where the length changes are measured

to get what the problem is. I find this confusing and of little help to see the relevant points.

We removed this paragraph, and included in line 105 the information that increasing debris cover is noted also during field surveys.

L71: I see Schnapfenkuchl M very well in Fig. 2, why do you write that it cannot be identified

at first glance?

As stated above, this should be the two Schnapfenkuchl glaciers V and H.

L72/73: What is meant with 'the geomorphological structure' and how would it look like when

it is 'dominated by ice dynamics'?

Alpine Glaciers with significant ice flow often show a change of aspect and and increasing slope at the margins. As you suggested to include the 2 geomorphology papers, we refer to those.

And why could this be expected for 'debris-covered valley glaciers', which are in many cases just near-stagnant down-wasting ice masses?

Because these tend to have steep lateral moraines in contrast to flat glacier tongues, so that there is a change in slope and aspect between ice and periglacial area. This is not the case here.

Though a bit hidden by the red arrow in Fig. 2, I can well see flow structures of debris

bands on the surface of Schnapfenkuchl M.

The discussion is focusing on Schnapfenkuchl V and H, as Schnapfenkuchl M is a 'classical glacier'. This is made clearer now. This glacier is hardly covered by debris. Shear crevasses and Ogives may be visible, but not discussed in this context. As you suggested to include geomorphology papers in the discussion, we have to adapt what you might refer to as 'sloppy wording' to correctly refer to the syntax of this discipline. I cannot see a total debris cover on Schnapfenkuchl M, and no deformation of this debris cover.

L75 (Fig. 2): The viewing angle of the image is a bit unfortunate as glaciers H and V are hard

to see from this perspective. I suggest showing a close-up - if at all - as I do not get what

the purpose of this image is (e.g. compared to Fig. 4). Moreover, 'rock glacier' is difficult

to read and the left one is also covering steep rock walls. Please do not present 'M' as a

glacier with increased debris cover and rarely exposed bare ice when the glacier is largely

free of any debris. Please also be precise with arrows, the remains of V are located

much higher up (where the snow fields are).

I agree that glaciers V and H and hard to see, but not because of view angle, but debris cover. We include close ups though.

Ok, I had the impression that there is a glacier mouth right at the location of the arrow visible in the orthofoto of 2015, with subsidence indicated in the elevation change map. Of course this can be also classified as dead ice.

L80: Show where this subset is located in Figs. 1 or 2 or 4.

We added this reference.

L81: Aren't these (deformed) 'stratigraphic layers' a sign of ice dynamics (see Leigh et al.

2019) and thus in contrast to the statement in L73?

I do not see a contradiction, as this layers located at the upper end of the glacier very likely indicate past accumulation patterns (see e.g. Hess, 1904). So far, we rarely observe deformation related ogives close to the summit, such features usually occur at glacier tongues.

L84 (Fig. 4): I find the white lines disturbing and would remove them, similarly for the rock

glacier outlines.

We removed the contour lines of elevation and redid the Figure.

 The multi-temporal glacier outlines are a bit too thin.

We adjusted the line style.

 Why is the glacier

name annotation not shown in the 2015 example?

No specific reasons.

Why do the image dates not match to

the outline dates?

As the mentioned in the method and the discussion of this study shows, it is hardly feasible to determine glacier outlines without the required high resolution surface elevation information in particular for small glacier. Thus, reasonable glacier inventories are in line with the ALS surveys and not with orthophotos, so that dates of orthophotos and outlines differ.

As it is nearly impossible to trace the glacier changes from image to

image, I would show only one image (maybe the first or the last one) and show it big.

Done

Please also note, there is lots of glacier free area to the left of V, but Larainglacier is partly

cut away. I suggest shifting the cropping to the east.

Thank you for this comment. We adjusted the cropping accordingly.

L85: 'show that the glacier changed quite rapidly within this time': this is hard to see from the

images but also difficult to see from the outlines (as the white and orange ones interfere).

Thank you for this hint, we changed that!

L86: 'The former accumulation area of Larainferner lost contact ... and 2009': Where is this

'former accumulation area? Please mark it (and the disconnection) to make the case.

Ok, marked.

Please also note that what has been disconnected was a tributary glacier. As one can

see from the images, snow accumulation still takes place below (and to the left and right)

of this separated glacier. So the statement seems incorrect.

Ok, we changed the text to 'from the main accumulation area'.

L87: Is 'eastern Schnapfenkuchl' H? Please show close-ups, to show the temporal evolution.

Yes. We show close ups now.

L88: 2019? The image is from 2015 and the outlines are from 2018.

Sorry, we corrected the typo.

L92/93: Figure 4 does not show an image from 2006 and most of the features in the list are neither visible nor marked on the images. As mass balances have not been measured (?), the term 'accumulation areas' seems misleading and should be replaced with 'remaining snow cover' or similar. I suggest removing the englacial drainage systems from the list and consult Leigh et al. (2019) for more suitable indicators (e.g. deformed debris bands). Maybe show a close-up with H M V only to see something?

Ok, we changed the Figure so it should be clear to which kind of data and date we refer to. Close up included.

L95: I do not understand this, M is still a well visible larger glacier. Writing here that you would hardly map any glaciers here is highly confusing and gives the (likely wrong) impression that you have no idea how a glacier looks like, i.e. it crashes the entire study!

That is correct, revised this. Schnapfenkuchl V and H are nice examples for the discussion raised in this study in contrast to the more classical glacier Schnapfenkuchl M.

L96 (Fig. 5): The order of images is strange, why not clockwise or 2006 and 2012 side-byside?

We skipped the Figure to include the close up suggested.

Why are there only arrows in 2006 and 2018, and where do they point to? What about marking possible glacier extents to reveal interpretation problems? As for Fig. 4, why is the glacier to the right partly cut?

The Figure was replaced by the orthophoto time series of Schnapfenkuchl H, as this time series could illustrate the problem the article is focusing on, and explain why it could be possible to identify glaciers in their current state.

L98: Why plural, there is only one flat summit glacier? Why 'completely disintegrates', it is still a single entity that has just lost some area?

The Figure is not part of the main text any more, the caption in the supplement Figure was rephrased.

Where do the 'melt rates of 1.5 m' come

from?

Measurements on 3 ablation stakes located on Jamtalferner, Fischer et al., 2016c-> cited now.

Why plural? Has a melt rate been measured at several places and is the mean value?

Yes.

Where have the melt rates been measured? Is it an annual rate?

Close to Jamspitze, Chalausspitze and Ochsenjoch. Yes, it is an annual rate.

L99: I see the developing rock outcrop but where is the debris cover? Where should the debris

come from when this is a 'summit glacier'?

The debris originates from the rocks at the summit.

[Figure]

[Figure]

[Figure]

I actually would call this development visible in the orthophotos 2015, 2018 and 2020 a disintegration, as the rock outcrop divides the glacier in at least two entities. But we can agree on downwaste.

L105: This seems to be incorrect. I can still see bare ice in 2012 and 2015

Even if bare ice is visible, the by far larger part is covered by debris at the glacier tongue. We improved the description, so that it is clear that we refer to the glacier tongue and glacier terminus here.

 (what is with

2018, is the ice back then?).

The main problem here is that Fig. 5 does not reveal where

the length change measurements are taken or which parts of the glacier turn from bare

ice to debris-covered ice.

Apart from this, I disagree with the interpretation by the authors:

There was likely some ice left in ice cored moraines in 2012, but by 2018 all ice melted.

We still have subsidence in the area.  So far, we have no indication that the ice melted completely in 2018.

So this is not really a glacier showing an increasingly debris-covered tongue, but glacier

ice that melted, leaving behind bare (debris covered) ground.

The debris on this glacier tongue mainly originates from englacial debris accumulating at the glacier surface while ice downwasting. This reliable finding is based on field trips to this area. However, if you have evidence on your thesis from field visit or literature we are open to that discussion.

L106: As this glacier has barely any debris cover, the impact on the accuracy is likely small.

In any case, a comparison to satellite-derived extents from 2016 would have been worthwhile

to make the point.

We included Sentinel-2 data in the revised version, but switched to Schnapfenkuchl H for all the explanations. It should be easier for the reader to follow the basic concept if we refer to a single glacier only.

L107: In the caption I read 1.5 m? #

Yes, more than one meter is no contradiction to 1.5 m and accounts for the range within the measurements.

Does the 'as measured on Jamtalferner' belong to the

'more than 1 m w.e.'? Why is it here?

As above, same stakes.

Now it reads as if the disintegration of the flat glacier

has been measured on Jamtalferner, which I find confusing.

We rephrased the sentence. This process was observed on Jamtalferner also.

L107: As stated above, this is likely a down-wasting but not (yet) a disintegrating glacier that

should be composed of several individual pieces. Moreover, disintegration has only a limited

impacted on mapping accuracy, so this example does not fit very well.

We switched to a presentation of Schnapfenkuchl H, as above. Please feel free to make up your opinion independent from the paper with the time series included in this response.

L108: Should this mean 'debris from rock fall'? Where should the debris come from during

disintegration or down-wasting? Is this all englacial debris?

We so far did not find a way to tackle the source, it seems that also subglacial sediments are transported to the surface by melt water if the ice thickness is lower that about 5 m.

L110: Why 'we will …in the coming decades'? This sounds as if this might happen at some

point in the future. Actually, we follow it for more than 3 decades now (Figs, 4, 5 6b, 9a).

… even for 150 years I would say. But this is the first time after 1855-1870 that a larger portion of glaciers just fade away…. The sentence was rephrased.

L110: What is a 'large glacier system'?

We rephrased the sentence.

L111: 'for the last hundred years': Again, a rather imprecise statement considering the well

documented advance phases of the glaciers over this time period. The related moraines

from the 1920s and 1980s advance can also be seen on some of the orthophotos.

In general, the glacier area was reduced between 1850 and 2006, despite the two smaller advances.

Similarly,

this is not only observed in the Silvretta region but in the entire Alps (and globally).

Yes, we agree, as stated in the first two paragraphs in the introduction.

L110-112: I would like to use the above example for pointing to the often confusing / interrupted

/ hard to grasp structure of the text. Why first writing that rapid recession will likely

occur in the future and then stating that it has already happened over the past 100 years?

We rephrased the sentence to clearly distinguish between the moderate recession of the fast and
the extreme downwaste of the last decade.

The first statement implies that it has not happened yet so the next sentence is a direct

confirmation that the first statement is wrong. Why not write that widespread glacier recession

was observed in the Alps over the past 150 years (incl. Silvretta), that recession

has more recently accelerated (this would require some proof of evidence) and will very

likely continue in the future (e.g. due to the committed mass loss)?

Thank you for this suggestion! We added the citations as proof for evidence, at least there are about
200 pages of reports available!

L 113: The 5 pages before only discuss aerial photography, the advantages of Lidar are not

discussed at all. The possibility to detect hidden ice by its down-wasting is not even mentioned.

Ok, thank you for this hint! We improved the description inline 60 (new line numbers), where LiDAR
appears first.

There is also nothing about the spatial resolution requirements to detect the relevant

details.

Spatial resolution is a topic discussed in the methods section. The third sentence in the Abstract
should also give some fist impression.

 In other words, the introduction discusses a lot of things but has a limited

perspective on what is relevant or motivating the study. Please introduce all relevant topics

appropriately, this is in my opinion the purpose of an introduction section.

Ok, we restructured the introduction and changed the Figures as suggested.

L113: Where is the 'glacier fade out' by Lidar been shown? I only see two distinct maps for different regions and a fixed time interval. 'Fade out' is for me a process occurring over a longer time and should thus be documented by time series. This has been done for outlines, but not for Lidar as stated here. Please check the text against the work presented.

Ok, we attached images to the summarizing graph.

L114/5: This statement is correct but what has it to do with the text before, e.g. the difficult mapping? What is the relevance of the glaciers presented in Figs. 2 - 5 for sea-level rise?

That is correct, we rephrased the text

L115: Instead of using glacial and post-glacial (which more refer to LGM glacier extents) I suggest writing 'a landscape with glaciers evolving into a glacier-free landscape'.

changed

L116: How does the monitoring by remote sensing help in 'dating of paleoglacial landforms'?

For example, by constraining the time period of succession in the periglacial areas, which is needed for the interpretation of pollen profiles and biodiversity records.

L118: I think this is not the definition of a glacier inventory.

We will change that to 'Austrian glacier inventory' and replace 'definition'by 'aim'.

L118: 'of all glaciers in the region': I do not find a figure showing this

Figure 1? Changed to ' in a specific region'

 and also Table 5 is only

presenting glacier areas from two points in time rather than discussing the changes.

Table 5 was shifted to the supplement. Change mapping also includes changes between two points in time. We included some time lapses and more time series to illustrate the processes.

L119/20: These are indeed two relevant questions, but where has this been shown in the study? Which landforms can be neglected and what is the impact?

Example added, I am sorry to say that we cannot provide a reliable global estimate with this small scale study as a first glimpse on new small scale processes.

L120: 'in a stage of rapid glacier shrinkage'

We wrote that 5 lines above already, following your suggestion there.

L 126 (bullet II): I miss a critical discussion if the short 2002 to 2004/06 interval makes sense at all, e.g. compared to uncertainties.

I agree on the question of the short period, but think that this is rather a topic for the discussion and not the introduction, where it is still included. The focus of the paper is the first repeat LiDAR inventory, older volume/area change data is just compared. This data exists, and there is no need to analyse the data of this short period. Nevertheless, as the extreme summer of 2003 is included, it is still interesting to discuss this short period.

There is also no image showing a dhdt map over

this time period.

Without such a proper documentation and discussion, the presented results

for this period seem unreliable and should thus not be presented. If the authors think

the comparison is reliable and reveals something important, please show it and discuss it!

As we have a number of additional task requested in this review, we decided for including the remote sensing inventories rather than reproduced the already published data.

L128: 'under conditions of rapid glacier decay'

Ok, we changed the sentence to 'of rapid glacier decay close to complete loss of ice'.

L131: As suggested in the general comments, please use a separate section to describe the

datasets and the methods. Please consider discussing in the introduction only the larger

background and motivation for the study, before presenting details of the study region in a

further section. And please keep out the results (Figs. 4 / 5) from the introduction!

Ok, the glacier margins 2017/2018 were removed from the Figs 4 and 5. We introduced a section of data and one on methods.

L133-136: All these datasets are shown in Section 1 before they are introduced …

It is not the datasets shown, but the development until our study took place. We tried to make that clear.

L169 (Fig. 6A): The bluish parts in the background (with positive values) look like a shaded

relief, indicating that the two DEMs used here have not been correctly co-registered.

We described the coregistration process with all uncertainties in detail, including the difference in resolution of the images. The deviations in the image just show these uncertainties well known and described. The differences result from the different spatial resolution of the data.

What does N, M, S mean?

Nord/North (N) Mitte/Middle (M) Süd/South (S)

It is unclear to me why the outlines have been mapped where

they are, what are the rules? I can follow the 'maximum change' rule near the terminus,

but partly the line is outside the bluer regions.

We changed the example Figure and added a more descriptive text.

L169 (Fig. 6B) I wonder why this figure requires outlines from 4 further dates, hiding details

of what is visible in the image? When the purpose is to reveal the accuracy of the dhdt interpretation

in Fig. 6A, I would show close-ups, compare it against outlines derived from

the orthophoto, and discuss differences in interpretation at the pixel level. The current

overlay tells me nothing in this regard.

We removed the older inventory shapes from the new descriptive Figure.

L180-250: This is all rather theoretical and should be there, but I am unclear how it helps

interpreting the results or how this information is supporting the goals of the study. I have

described above and in the general comments what I would have liked to see here (e.g. a

comparison of the results obtained with different methods or how the differences in interpretation

impact on the overall results).

This part is important, as it explains why you see the bluish color above. I consider the information on accuracy as a major point in every scientific study. We additionally included the data and Figures you requested, so that every need should be satisfied now.

L220: Shouldn't this be Section 2.5?

Yes, but the numbering is changed now by splitting the data and methods section into 'data' and 'methods'.

L250-290: The results section is only presenting inventory data and volume changes. The

results for the two research questions (L119/20) are missing. There is a qualitative discussion

in Section 4, but no quantitative evidence is given to support the statements.

This is part of the discussion, and then found in the conclusion. It is also stated in the introduction that here we only reach the level of discussion. I cannot confirm that no evidence is given, as numerous data is presented. It is not entirely clear to me what you would expect, but we tried to clarify our text.

L285: Why should there be a relation between mass balance and glacier area?

Basically because b= B/A, here you definitely find the Area A in an equation. Please see the number of recent publications in the influence of glacier area on mass balance.

With respect to your question, small glaciers are situated in specific settings.

We have been interested to know how the specific glacier of different sizes reacted to climate change. We do not propose a relation, it is just that the last inventories (in Austria as well as Switzerland) showed different responses of small and large glaciers, mainly because the tongues of the large glaciers are subject to large ablation rates.

 Wouldn't be

glacier mean slope, aspect or elevation the more interesting variables for a scatter plot?

We found that the sample size is too small for such a comparison.

L288: Please sort the table alphabetically or refer to the numbers in Fig. 1. Mark ALL glaciers

listed in this table in Fig. 1 (rather than numbering a selection that is never used for

anything). This Table might be shown in the supplement.

I am sorry to say that it would be too much to have 46 labels in Figure 1. Please refer to the data
available at Pangaea for this information.

L306: I think ponds or crevasses are not 'stable surface structures'.

We skipped the word 'stable'. Generally, stability refers to certain period of time, we all know that
glaciers basically are less stable than solid rock.

L308: I do not understand this connection, what does a bore hole tell me about surface velocities

due to sliding debris and rock?

The vertical profile of velocity and the stratigraphy allows to see which stratigraphic layers of the
rock glaciers move relative to others

And why 'sliding' on the ice?

We refer here to the deformation profile of Lazaun rock glacier, Krainer et al- see the draft.

Isn't a rock glacier a

frozen body of ice and rock (below zero degrees) that is creeping?

Yes. However, not always rock and ice can be considered to be evenly distributed within the RG.

L309: Should this mean exits or emerges at the terminus? Does this emerging water define

where the terminus is?

It states 'exists', as not for every structure runoff exists or comes to the surface. At least we take the
samples where we observe the current outlet of the ice body.

L315: This might be correct but what about the effort and the limits imposed by spatial resolution?

When resources are limited and aerial photography or Lidar DEMs are unavailable,

one has to decide what to do with the available personnel resources and datasets to

get an inventory finalized. I would have expected here a recommendation about such limits

from the datasets analysed here.

There is a clear if, 'if we want to keep track'. Limits are found a bit further in the text, and as you
suggested we can include close ups so that it is easier for the reader to see the limitations.

L316: This should be Section 4.2 rather than 4.4 and so forth for 4.5 (=> 4.3), etc.

Yes, as above.

L322: How can ice melt in a region with permafrost, i.e. with temperatures below zero?

Why not? We often observe permafrost in a suited microclimatic setting neighboring melting snow and ice or even running water. We observe daily, seasonal variability of temperature, and about 70% of the energy for melt stems from radiation!

L323: Why debris flows? This looks as ice that is exposed because rocks slid down on a

steep slope (cf. Fig. 3).

There are no rocks to find in this location, rather small grain sizes. This often happens during heavy precipitation events. Nevertheless, debris flow (not in a hydrological sense) may be the wrong wording here, changed to debris sliding down the ice.

L325: 'needs a high temporal frequency of inventory data' I think this issue cannot be resolved

by just increasing the frequency of inventory data. The temporal and spatial scale

we are discussing here points more to a frequent (daily?) observation by a good webcam

with repeat DEMs derived from drones.

Sorry we have no data to compare to the annual or even subseasonal time scale. We argue that at least every 5 to 10 years a high resolution inventory will be needed to keep track. We will be happy to discuss a study and a paper on cams and drones!

 Such an effort can likely be justified for scientific

investigations, but for regular monitoring it seems to be too demanding.

This may depend on the hazard potential and the corresponding area of interest. This study is focusing on basic science.

L334: The inserts show some bare ice but where are the debris flows?

We replaced the Figure, as obviously more close ups are needed to follow.

L340ff: What about the remote sensing based glacier definition from GLIMS?

See above, will be included.

L361: Just that two groups of scientists interpreted the landscape differently does not mean

that a transformation of the mapped forms has taken place.

Correct, and try to state here that it is a different interpretation of the same landform.

In a region with permafrost

and steep slopes the surface can also start creeping without the ice from a former glacier.

correct

Wouldn't also the glaciers temperature regime require a change from sliding to creeping?

Up to now there are no thermistor chains available at the tongues, nor measurements of sliding velocities.

L386. What about comparing mass balances to Silvretta Glacier?

We tried to focus on the Austrian inventory. The comparison to the mass balances of Silvretta glacier and Jamtal Ferner would open another topic and will be subject to a paper focusing on the direct mass balance series.

L424: Why is Fig. 10 not in the methods section?

Because we consider this Figure to be a result and a graphical summary. We added close ups to make that point clearer, and improved the description of methods.

Response to RC #3 from 16.07. 2021 https://tc.copernicus.org/preprints/tc-2020-376/

We thank reviewer #3 for the comments and suggestions which will help us to improve the manuscript!

Review in black letters

Response in blue letters

This manuscript is an interesting and well written paper about the evolution of small-scale glaciers in a sub-region of the European Alps. Disintegration and especially steady covering by rock debris creates problems in mapping glaciers and creating glacier inventories.

The authors describe very well the material and methods and have a thorough discussion at the end. Interesting is the assumption of an updated definition of the term glacier.

What about glacier movement in the definition? Can that be discarded?

> We thank the reviewer for their interest! The discussion on existing ice dynamics is important, as stagnant ice bodies of larger sizes can only exist as transient states (and will melt soon under current conditions if not covered by thick debris layers). We showed that the increase or decrease of ice flow velocity is one of the earliest responding indicator of glacier state (Stocker Waldhuber et al., 2019 https://essd.copernicus.org/articles/11/705/2019/). Including dynamics in a definition would mean to exclude stagnant ice bodies, and include them again if they start to move again (which can happen within a season). As in the moment we cannot measure dynamics of a buried ice body without drilling, all arguments point towards discarding the ice movement in the definition of a glacier, with the result of including various types of stagnant and buried ice.

Minor comments:

L254: … the latest period was 2.4%, which is … (you wrote loss; therefore, it should be a positive number)

> Changed

Table 3: should be -17km² area change in 1969

> Thank you for pointing out this typo, we changed that.

L266ff (and further in the text): please check terms highest/lowest/maximum etc. à they should all be the other way round as you refer to negative numbers

> As for L254, changed.

L270f: …, reducing the overall volume loss as no ice to melt is left in the areas with highest ablations in the past.--> this sentence is not clear to me.

> This sentence is definitely too long and was be rephrased. At Jamtalferner, the areas at the glacier tongues where more that 6 m ice ablation was measured are ice free now. Now only the areas with about 4 m of ablation in the past are left to contribute to volume change. Therefore, despite an increase in ablation at individual stakes, total volume loss decreased.

Table 5: I would suggest a map with different colors/symbols instead of the table.

> As one of the previous reviewers suggested to move Table 5 to the supplements, we will be happy to add a map as suggested in the main text. The main results are found in Figure 8.

L319ff: Could another reason be a change in debris cover?

> We mentioned that changes in debris cover can also induce changes in surface elevation.

L375: does away with à better use overcomes?

> rephrased.

L391: Caucasus comparable to Silvretta?

> We included a comment on the difference of these mountain regions, sorry not to have mentioned that, as it seemed very clear to us.

L419: … the future

> changed

Fig. 10: I do not understand how you discriminate between volume change? and debris accumulation dominant? following the thickness change. There are two arrows without further indication.

> Thank you for pointing out these two issues. There is only one experiment regarding debris accumulation on ice (with the help of net) on a single location where debris comes from the rock faces, so that in the moment we see no way to estimate the exact amount of debris accumulation. The other problem is that there is no way to distinguish 'external' debris from englacial debris coming to the surface. Here we are not sure how to design an observation procedure.

> We restructured the Figure and added visuals so that it should be easier to follow.

L460: The glacier inventory data is stored in https://doi.org/10.1594/PANGAEA.844988.

**Citation**: https://doi.org/10.5194/tc-2020-376-RC3

> We changed that and included the citation

---

## Author Response (AR2)

Dear authors,

Thanks for addressing all the referees' comments, questions and suggestions. Your manuscript really has been revised significantly. I also think that it has improved.

I have read through the revised manuscript in detail and there is still quite a lot of editing to be done. Please see my comments and suggestions below. Besides these I encourage you to carefully read through the manuscript again for additional inconsistencies - I know that many of these are probably due to the amount of editing you have done based on the reviews.

Best,
Louise Sandberg Sørensen

Dear Luise Sandberg Sørensen,
Thank you so much for your comments and suggestions. We went through the manuscript and corrected as suggested.
We hope that the captured all the typos now, we also found some typos in Figures and missing citations/references. The search for additional typos was carried out in the changes accepted version, as I experienced that some of the typos are visible in the change track version.
Kind regards, Andrea Fischer
* * *
The abstract is written in a passive voice which makes it difficult as a reader so understand what has been done by you and what has been done previously by others. I would suggest that you revise the abstract to be in an active voice.
We rewrote the abstract accordingly.

L 18-19: - 0.6 -> -0.6
changed

L 21: -0.8 m ±0.1 w.e./year -> -0.8 ±0.1 m w.e./year
changed

L 30: I don't think that mapping glaciers is necessary to *tackle* climate change. Maybe *assessing* is a better word?
Yes indeed, changed.

L 39: for example debris-covered -> for example mapping of debris-covered
changed

L 43: enhanced -> improved
changed

L 48: I don't think that you have defined DEM at this point. Also, which of the DEMs you mentioned here is the one outlined with the black line in figure 1? This is not clear to me. Maybe add that information in the figure caption.
DEM explained, sentence rephrased to:
Now a repeat federal LiDAR DEM is available for several regions in Austria, as the Silvretta range where the DEM Vorarlberg dates from 2017 and the DEM of theTyrolean Silvretta from 2018 (black polygon in Figure 1).
Information to caption added: The DEM of the Tyrolean part dates from 2018 (black outline), of Vorarlberg from 2017.

L 58: Repeat LiDAR -> repeated LiDAR measurements.
        tackle -> map

changed

L 59: .. since then. Since when? 2003?
Yes, changed to: since 2003

L 61-62: I did not understand this sentence. Would the suggested change make sense?:
The evolution of the other glaciers in the region is similar, so that it was necessary to include the interpretation of volume changes to delineate the debris-covered glacier margins. -> The evolution of the other glaciers in the region is similar, making it useful include the derived volume changes in the analysis to delineate the debris-covered glacier margins.

Yes, this is exactly what we mean -> changed.

L 65: Heavy -> Large
changed
     Is topology the right term here? I would use the term topography.
What we refer to is the disintegration of larger glaciers into smaller ones, in terms of polygon outlines a change in topology. As these is mixing up real world and GIS, we follow your suggestion and stay in real world-> using 'topography'.

L 69: lager -> larger
changed

Figure 2: What do the red arrows on the figure indicate?
Exposed ice in the close ups-> caption changed.

Figure 4 caption: Bare ice exposed at Schnapfenkuchl glacier V in the orthophoto of 2015 (red arrows), with stratigraphic layers (yellow arrows) indicating sedimentary ice or firn. ->
Bare ice exposed (red arrows) at Schnapfenkuchl glacier V in an orthophoto from 2015 (red arrows), with stratigraphic layers (yellow arrows) indicating sedimentary ice or firn.

L 86: Is it *classical* or *classic* glacier? And could you briefly define what a classic(al?) glacier is?
Also, in the following sections you use simply 'glacier'. Should this also be classic glacier?
It seems to be the clearest and most appropriate phrasing to just use the term 'glacier' instead of 'classic' or 'classical glacier'. Changed.

L 87: If I understand correctly, I would suggest the following edit:
Schnapfenkuchl H glacier can be clearly identified as glaciers, while in 2009, 2015 and 2020 we would hardly map Schnapfenkuchl V and H as glaciers compiling the first glacier inventory of the region -> the Schnapfenkuchl H glacier can be clearly identified as a glacier, while in 2009, 2015 and 2020 we would hardly map Schnapfenkuchl V and H as glaciers compiling the first glacier inventory of the region, due to the debris cover.
changed
L 94: What does *this process* refer to?
Rephrased so that it should be clearer that we discuss the evolution of the debris cover.

L 96: what do you mean by *former* firn area?
Where prior to 2003 the firn cover was located. We skipped that part of the sentence, as there is no need to discuss which parts of the glacier are NOT covered by debris here.

L 97: w.e. per year?
Yes, changed.

L 96-99: This whole section is unclear to me. How exactly are the small glaciers in previous inventories affected by the high malt rates? Please rephrase and clarify, as this seems to be an important point.
The section is rephrased to: tThe small glaciers which remained fairly unchanged in the last inventories (Abermann et al., 2009) were affected in a way which is relevant for compiling glacier inventories: the formerly

ice covered and now exposed steep rock faces release rock and debris onto the glacier surface. Melt rates of more than 1 m w.e. per year recorded at the Jamtalferner even in elevations above 3000 m (Fischer et al., 2016c) were recorded after 2003, removing the firn layers and decrease albedo. The ice thickness in higher elevations is often smaller than at the glacier tongue, that an annual melt rate of 1 m could correspond to 5% of the total thickness (e.g. Fischer and Kuhn, 2013), exposing rock outcrops of a rough bed topography..

L 106-107:         This is important on a local and global level, as even small glaciers contribute to sea level rise (Bahr and Radic, 2012) and can be significant for local hydrological and hazard management. -> This is important on a local and global level, as small glaciers not only contribute to sea level rise (Bahr and Radic, 2012) but can also be important for local hydrological and hazard management.
changed

L 110: are a perfect test site for analysing -> are perfect for testing and analysing
changed

L 113: aims at tackling -> can be used to estimate
changed

L114: This raises the research questions which of the potentially transient cryogenic structures (e.g. Figure 5, year 2020) should remain part of a glacier inventory, and what the effect of neglecting ice remnants on inventory data would be. ->
 This raises the following research questions: Which of the potentially transient cryogenic structures (e.g. Figure 5, year 2020) should remain part of a glacier inventory, and what is the possible effect of neglecting ice remnants in the inventory.
changed

L 118: presents -> presents:
changed

Figure 5 caption: again, is it *classical* glacier?
Changed to 'glacier'

L 136: Spatial -> The Spatial
changed

L 137: You should define DEM when you use it first – not here.
Changed, skipped here.

L 141: What do you want to say bt by stating that coregistration is not considered state of the art? If it is not, why do you do it (if you did – which is not clear)?
The sentence says that the coregistration of the full waveform DEM is not state of the art, as point clouds and algorithms referring not the all points in the same way are used. The sentence is rephrased.

L 143: what is a pass area?
Changed to control area: flat areas with the exact elevation known from DGPS surveys.

L 150: changes to the LIA -> changes relative to the LIA
        Did you define LIA?
Changed & defined

L 156-157: Is this the same information as you provide in lines 146-147? If so, delete this text but move the reference to the right place
No, here the accuracy of previous DEMs is described, above the uncertainty of 2017/18 DEMs is given.

L 163: What is meant by *intended* point densities?
We skip the 'intended' here, as it is obvious that the point density also depends on surface topography, and so the effective point density on a rough surface can differ from the one measured at a flat ground.

L 164: infinitesimally accurate 'real surface' -> actual surface
changed

L 166: What does the 'nevertheless' contrast?
We skipped the 'nevertheless'.
L 173: not only at *classic* glaciers?
We now stick to 'glaciers' instead of 'classic glaciers'.

L 177: It seems to me that the statistics (0 +/- 0.6m) you mention here is related to the buffer zone (1000-2000m) in table 2 . If that is correct, why do you write that it is related to the unstable zone (1000-2000m). Or do you simply mean:
"In the 1000-2000 m buffer, we found a mean elevation difference of 0.0±0.6 m."?
Yes, the meaning was that simple, changed.

L 185-186: Studies on the derivation of DEMs from LiDAR point clouds reveal that a slope steeper than about 40° potentially exhibits larger deviation from the 'true' surface (Sailer et al., 2014) -> Studies on the derivation of DEMs from LiDAR point clouds reveal that at slopes steeper than about 40°, it potentially exhibits larger deviations from the actual surface (Sailer et al., 2014)
changed

L 191: Silvretta. There -> Silvretta, where 90       changed
      Presents -> has          changed

Figure 6 caption: slopes for glaciers -> slopes for classic glaciers?
We now stick to the term 'glaciers'

L 201: The glacier outlines were mapped… -> We have mapped glacier outlines…
changed

Figure 7 caption: Could you better explain what the arrows 1 and 2 shows?
The position of the ice margin, changed

Methods: one basic question: did you map the glacier outlines by combining the orthophotos, the DEM hillshade and the thickness change? From the figure 7 caption text it seems that you do it separately? Also, would it make sense to show the actual mapped margin in figure 7 as you write that this is what you do?
The orthophoto is just shown for orientation, as it is not from the same year. We use it just for comparison/plausibility check. We rephrased the caption.

L 206: Did you define VIS?
As we use VIS just once, we spelled it out.

L 217: plausibility -> consistency?
changed

L 218: What is meant by a minimum size threshold? That they have only mapped glaciers of a given minimum size? In that case, what is that minimum size in previous inventories?
We changed the wording so that it is clear that the Austrian glacier inventories generally did not apply a minimum size criteria.

L 223: 1 x 1 m^2 -> (1 m x 1 m)
changed

L 224: format t1 and t2 correctly
done

L 228: uncertainty in area -> uncertainty in the glaciated area?
Changed to glacier covered area

L 230: did you define $\Delta A$?
done

L 234: from volume -> from the volume
Changed

L 241: When you write that the firn layer has melted do you mean that is has completely disappeared or that it has decreased in thickness/extent?
Rephrased to 'melted completely'.

L 242-244: It is not clear to me why a slower ice velocity leads to more cavities. Is this due to more stable hydrological channels in the ice?
Yes.

Equations: please go through the manuscript carefully to ensure that you have the same formatting of variables in the text as they have in the equations. These are inconsistent in several places.
Done

L 261-262: Please specify the periods that you refer to here as yyyy-yyyy.
Changed

       area 0.303 -> area of 0.303
changed

L 263: visible, mean -> visible, and mean
changed

L 265: What does it mean that the *ice merely melted*?
Correct, ice does melt or not, we skipped 'merely'.

L 277: Here you use the term specific direct mass balance. How does this differ from the specific mass balance used previously?
It is the specific mass balance, we skipped the emphasis that this is a directly measured in situ balance.

Table 4: Would it be a good idea to briefly explain in the main text the difference between the different mass balances (*geodetic, specific geodetic, annual specific geodetic*) and why it's interesting to show all results?
Also, please go through the manuscript and ensure that you use a consistent terminology wrt mass balances. You use many different terms – and I am not sure if some of them are actually the same or not; besides those just mentioned you also use *specific mass balance* and *specific direct mass balance.*
*As there are so many papers on comparing direct and geodetic mass balances, we actually do not want to open this floor, as this is an extra topic. To explain what actually lead to the difficulties in mapping glacier, we consider it helpful to briefly mention the directly measured mass loss. The geodetic mass balances are defined in section 3.3.*

L 292: delete *extremely*
done

L 293: how can a stable location be due to loss of ice? Do you mean disappearance of ice / retreat of ice cover?
Rephrased to 'total loss of ablation area'.

L 296: significant is a word with statistical meaning. Unless you have actually checked this, I suggest you change it to *high*
*We skipped 'significant', as we quantify the impact later on.*

L 298: what balance do you refer to here?
We added 'geodetic'.

L 300: mass balance -> mass balance estimates
changed

L 299: maybe it's due to my limited knowledge on permafrost, but why would you consider to include this in a glacier inventory?
I suppose this refers to L 309; We could want to include all the ice remnants of a glacier in an inventory, to avoid very complicated definitions what to include and what to exclude.

Figure 9 caption: volume change -> elevation change
done
          20062018 -> 2006-2018
done

L 346: Please proved references for those inventories you mention here
(Paul et al., 2020) added
L 354: The purpose of the sentence "This the area affected by volume change can be constrained" in unclear. What do you want to say here?
We skipped the sentence, it is clear from the Figure that we can delineate the area where ice was located at the beginning of the period.
          Actual presence -> actual current presence
changed

L 368: provide units on the 0.098
km² added

Page 21: I would suggest to move these citations to the supplementary material. It takes up a lot of space in the manuscript, and doesn't seem very important. Is the main point that the definitions have changes – but maybe not so much exactly how?
Done

L 405: Cogley et al. (2011) does not appear in the reference list. Please check that all references are there.
Done
L 411: please define with years the period you mention
Done

L 433: I would suggest to delete 'real world'
Done

L 450: but but -> but
changed
L 460: based <1m -> based on <1m
changed
Figure 12: I do not think that this figure is in the right place. It belongs in the methods section as it was also pointed out by one of the referees.
changed

---

## Author Response (AR3)

Authors response to the editor's comments

We thank the editor for pointing out two technical errors and corrected them.

L. 16: resolution The -> resolution. The

Corrected

L. 110: increase -> increasing

As line 110 is: 'decade, but sooner or later will in other mountain regions also, we have to validate and elaborate our monitoring strategies to', we searched for 'increase' and found it in lines 91 and 95, in both lines we replaced ' increase in debris…' by 'increasing debris'.